# Conserved nucleocytoplasmic density homeostasis drives cellular organization across eukaryotes

Abin Biswas [1,2,3,9], Omar Muñoz[3,4,9], Kyoohyun Kim [2,3,9], Carsten Hoege[5], Benjamin M. Lorton[6], Rainer Nikolay [7], Matthew L. Kraushar [7], David Shechter [6], Jochen Guck [2,3,4] ✉, Vasily Zaburdaev[3,4] ✉ & Simone Reber [1,8] ✉

The confinement of macromolecules has profound implications for cellular biochemistry. It generates environments with specific physical properties affecting diffusion, macromolecular crowding, and reaction rates. Yet, it remains unknown how intracellular density distributions emerge and affect cellular physiology. Here, we show that the nucleus is less dense than the cytoplasm and that living systems establish a conserved density ratio between these compartments due to a pressure balance across the nuclear envelope. Nuclear transport establishes a specific nuclear proteome that exerts a colloid osmotic pressure, which, assisted by chromatin pressure, increases nuclear volume. During *C. elegans* development, the nuclear-to-cytoplasmic *density* ratio is robustly maintained even when nuclear-to-cytoplasmic *volume* ratios change. We show that loss of density homeostasis correlates with altered cell functions like senescence and propose density distributions as key markers in pathophysiology. In summary, this study reveals a homeostatic coupling of macromolecular densities that drives cellular organization and function.

The cytoplasm and nucleoplasm of eukaryotic cells are complex aqueous solutions of macromolecules, small organic molecules, and ions. Both the cytoplasm and the nucleoplasm are active, self-organizing systems that exhibit cell-type and cell-cycle specific emergent properties[1,2]. At the mesoscale, emergent material properties include cytoplasmic viscoelasticity[3–6], contractility[7], and molecular crowdedness[8]. These material properties are dynamically intertwined with the spatiotemporal organization of the cell. For example, cytoplasmic viscoelasticity affects cytoskeletal organization and vice versa[9], bulk contractility affects the localization of organelles[10–12] and macromolecular crowding affects the formation of phase-separated droplets[13,14]. Despite the growing evidence that nuclear and

cytoplasmic properties are highly relevant for cellular physiology[2,15], we do not understand how their bulk material properties emerge and how they affect cellular organization.

One fundamental and defining material property of a cell's interior is its density (dry mass per unit volume). For a given cell type, cytoplasmic density is tightly controlled and loss of cell density homeostasis correlates with altered cell function and disease states[16–18]. So, while it is evident that changes in intracellular density have far-reaching consequences for cellular physiology, it remains unknown how these changes affect processes at the subcellular level. For example, the nucleus is usually considered as a strongly crowded environment[19,20], which harbors millions of protein complexes in

[1]Max Planck Institute for Infection Biology, Berlin, Germany. [2]Max Planck Institute for the Science of Light, Erlangen, Germany. [3]Max-Planck-Zentrum für Physik und Medizin, Erlangen, Germany. [4]Friedrich-Alexander-Universität Erlangen-Nürnberg (FAU), Erlangen, Germany. [5]Max Planck Institute of Molecular Cell Biology & Genetics, Dresden, Germany. [6]Albert Einstein College of Medicine, Bronx, NY, USA. [7]Max Planck Institute for Molecular Genetics, Berlin, Germany. [8]University of Applied Sciences Berlin, Berlin, Germany. [9]These authors contributed equally: Abin Biswas, Omar Muñoz, Kyoohyun Kim.
✉e-mail: jochen.guck@mpl.mpg.de; vasily.zaburdaev@fau.de; reber@mpiib-berlin.mpg.de

addition to the genome. Recent experimental evidence, however, shows that nuclei have a lower refractive index (RI) when compared to the cytoplasm, indicative of a lower density[21–23]. Cells keep the lower nuclear RI throughout the cell cycle, even when physically and chemically challenged[23]. This suggests that the lower nuclear density is robustly maintained by the cell. Only in some specialized cells or growth conditions nuclei have been reported to be denser than the cytoplasm[24–26]. Yet, it remains largely unknown how intracellular densities are established and maintained across cell compartments.

Just like cellular density, cell size is narrowly constrained within a specific cell type. Although it has recently been observed that cytoplasmic density decreases when cells grow too large[27], it remains unknown whether cell density and volume are mechanically linked. Cell size, however, is one of the critical parameters controlling the size of intracellular structures[28–30]. A well-known example is the constant nuclear-to-cytoplasmic (NC) volume ratio first described by Richard Hertwig as early as 1903[31]. Despite a century of size-scaling observations, the mechanisms coordinating the growth and size of subcellular structures remain largely elusive.

In this study, we show a homeostatic coupling of the nuclear volume to the cytoplasm by a pressure balance mechanism across the nuclear envelope to maintain a constant density ratio of $0.8 \pm 0.1$ between these two compartments. We discover this constant NC density ratio to be conserved in 10 model systems across the eukaryotic kingdom, suggesting that this measure is a fundamental cellular characteristic. To mechanistically decipher how nuclear density is set, we reconstitute nuclear assembly and growth in *Xenopus* egg extracts. We find that nuclear import establishes a colloid osmotic pressure, which − assisted by chromatin pressure − inflates the *Xenopus* nucleus to its observed volume and results in a nuclear density that is lower than that of the cytoplasm. The NC density ratio homeostasis breaks down during senescence suggesting that this ratio could serve as a marker for pathophysiological states. Mechanistically, the density inversion is due to perturbed ribosome biogenesis which dilutes the cytoplasm and lowers its density. In vitro reconstitution experiments confirm our observations from diseased cells and help elucidate how cytoplasmic density is set by heavy, osmotically-inactive components. We substantiate our in vitro findings by observations in early *C. elegans* embryos, where NC density ratios are robustly maintained throughout development. Based on general biophysical principles of pressure balance and kinetics of active transport, we present a unifying theoretical framework that robustly predicts nuclear volume and density and explains how these respond to environmental changes. In summary, we propose that the maintenance of a constant NC density ratio is the biophysical driver of the frequently observed scaling behavior of nuclear size with cell size.

## Results

### During assembly, nuclei establish a lower density than the surrounding cytoplasm

While it is known that overall cell density shows little variation within a given cell type[17], there is limited understanding on how density varies at the subcellular scale. To close this gap in knowledge, we used correlative fluorescence and optical diffraction tomography (ODT) to identify subcellular structures using fluorescence and measure their corresponding 3D refractive index in individual cells (Supplementary Fig. 1a–d)[22]. In biological materials, the RI is linearly proportional to the dry mass density ($\rho$) of constituent macromolecules[32–34]. Thus, these RI tomograms provide a direct, label-free, and quantitative readout of cellular mass and density distributions (for details see Materials and Methods). In a set of 9 model systems, ranging from yeast to human cells, we consistently found the nucleoplasm to be less dense than the cytoplasm (Fig. 1a). While the absolute cytoplasmic and nucleoplasmic densities differed between species (Fig. 1b, Supplementary Table 1), the ratio of nuclear-to-cytoplasmic density was invariantly $0.8 \pm 0.1$

(Fig. 1c), suggesting that this ratio is actively maintained by the cell. These observations imply that after mitosis, when the nucleoplasm and cytoplasm are fully mixed, the assembling nucleus needs to reestablish a lower density. The formation of eukaryotic nuclei upon mitotic exit therefore provides an experimental system to examine the mechanisms that control nuclear density. To this aim, we reconstituted nuclear assembly in vitro using *X. laevis* egg extracts. Such nuclei are known to recapitulate many essential processes such as DNA replication, nucleocytoplasmic transport, and nuclear growth[35]. We visualized chromatin, membranes, chromatin replication, and nuclear volume over time (Fig. 1d, e, Supplementary Fig. 1e, f). Additionally, we visualized and quantified nuclear import by the accumulation of green fluorescent protein (GFP) fused to a nuclear localization signal (NLS) (Fig. 1d, f). Simultaneously, we measured the RI tomograms and calculated nuclear dry mass, which gradually increased over time (Fig. 1d, RI, e, $M_n$). At 60 min, the average reconstituted *X. laevis* nucleus had a dry mass of $51 \pm 2$ pg. Combining volumetric and dry mass information allowed us to calculate the density of assembling and growing nuclei over time (Fig. 1g, Supplementary Fig. 1g). Once densely packed sperm chromatin ($\rho_{n\_0min} = 183 \pm 7$ mg/mL) was added to *Xenopus* egg extract, its density quickly decreased within five minutes ($\rho_{n\_5min} = 104 \pm 1$ mg/mL) to level with that of the cytoplasm ($\rho_{cyto} = 100 \pm 2$ mg/mL). Concurrent with the start of nuclear import, nuclear density further reduced and then steadily remained below the density of the cytoplasm while the nuclei continued to grow ($\rho_{n\_15min} = 97.6 \pm 0.9$ mg/mL, $\rho_{n\_30min} = 91.7 \pm 0.9$ mg/mL, $\rho_{n\_60min} = 91.4 \pm 0.8$ mg/mL). Thus, our correlative fluorescence and ODT set-up allowed us to systematically measure cytoplasmic and nuclear densities with high spatial (120 nm) and temporal (1 sec) resolution (Supplementary Fig. 1h, i). Taken together, we found that the *Xenopus* nucleus robustly establishes a lower density than the surrounding cytoplasm within 15 min of nuclear assembly. Such a constant NC density ratio of approximately 0.8 − ubiquitous to nine other species across the eukaryotic kingdom − suggests that this measure is a fundamental cellular characteristic. We next ventured to identify the key biochemical and biophysical mechanisms responsible for the observed dynamics in nuclear density.

### Nucleoplasmin-dependent chromatin decondensation and nuclear import lower nuclear density

The decrease in nuclear density as described above can be nominally divided into two phases. A first fast decrease in density that happens concomitantly with apparent sperm chromatin decondensation. This is followed by a second − more gradual − decrease in density which starts concurrently with nucleocytoplasmic transport and results in a nuclear density below that of the cytoplasm (Fig. 2a). What are the molecular processes responsible for the decrease in nuclear density? It is known that highly compact *Xenopus* sperm chromatin undergoes rapid Nucleoplasmin-dependent decondensation in *Xenopus* egg extracts. Nucleoplasmin (Npm2) is a pentameric embryonic histone chaperone that removes sperm-specific basic proteins, binds core histones, and promotes their assembly into nucleosomes[36–38]. To quantify how chromatin decondensation affects its density, we supplied *Xenopus* sperm chromatin (in buffer) with purified recombinant Npm2 (Supplementary Fig. 2a) and measured chromatin density over time: within 10 min, chromatin rapidly decondensed and increased in volume while its density decreased (Fig. 2b, c, Supplementary Fig. 2b). Next, to show that Npm2 is both necessary and sufficient for the initial reduction in nuclear density, we performed immunodepletion and add-back experiments (Fig. 2d). Consistent with previous reports[37], Npm2 depletion left the sperm chromatin condensed, and adding back Npm2 led to its quick decondensation (Fig. 2e, Supplementary Fig. 2c–e). In Npm2-depleted extracts, sperm chromatin had a higher density than in control-depleted extracts (Fig. 2f, $\rho_{10min} = 115 \pm 5$ mg/mL versus $98 \pm 5$ mg/mL), suggesting that Npm2-dependent decondensation was necessary for the initial drop in density. Sperm

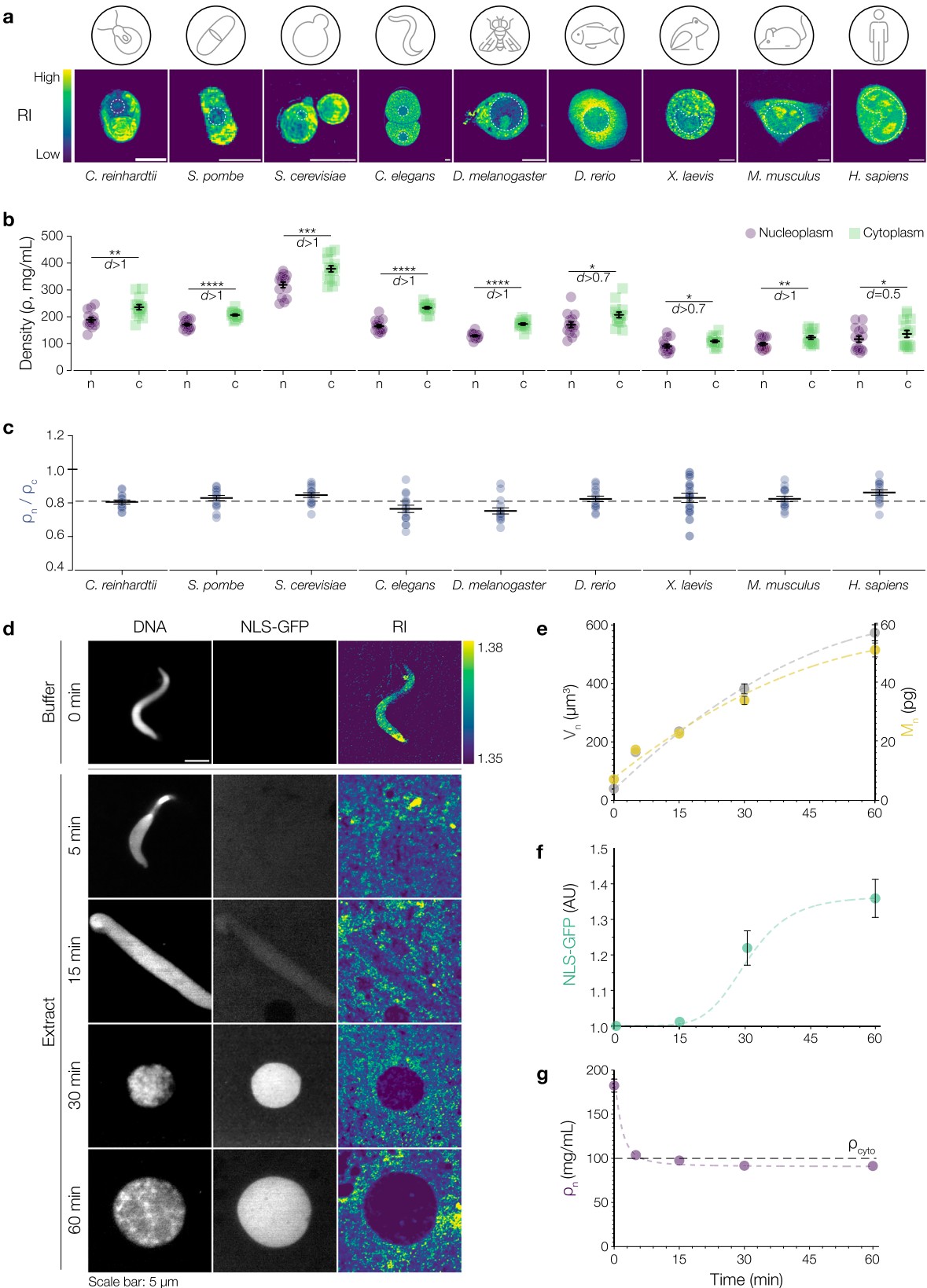

chromatin in control-depleted and Npm2-supplied extracts had a similar density as control nuclei (Fig. 2f, $\rho_{10\,min} = 98 \pm 5$ mg/mL versus $89 \pm 3$ mg/mL), suggesting that Npm2-dependent chromatin decondensation was sufficient for the initial reduction in nuclear density.

The further reduction in nuclear density below the density of the cytoplasm started concurrently with the accumulation of nuclear NLS-GFP suggesting that nucleocytoplasmic transport might play a role. Therefore, we assembled nuclei in the presence of import inhibitors (Fig. 3a)[39,40]. The successful inhibition of nuclear import was visualized by a significant reduction of NLS-GFP accumulation in the nucleus (Fig. 3a, b). Import-deficient nuclei assembled after chromatin addition and were initially similar in size. After 60 min, however, the import-

**Fig. 1 | During assembly, nuclei establish a lower density than the surrounding cytoplasm. a** Cells from 9 different model eukaryotes imaged via Optical Diffraction Tomography (ODT). Refractive index (RI) distribution from low (blue) to high (yellow). White dashed line indicates the location of the nuclei in the representative cells. **b** Absolute values of nucleoplasmic (purple) and cytoplasmic (light green) densities ($\rho$) vary within the different eukaryotic cells. Each symbol represents a measurement from one cell. **c** NC (nuclear-to-cytoplasmic) density is conserved across eukaryotes. Ratio of nuclear and cytoplasmic density ($\rho_n/\rho_c$, blue) across different eukaryotes. Each circle represents the NC density ratio of one cell. Nuclear densities were measured excluding nucleoli. Black dashed line shows the average NC density ratio of $0.8 \pm 0.1$. **d** Density reduces during nuclear assembly in *Xenopus* egg extracts. Representative images at different time points of nuclear assembly. Fluorescence images of DNA (left panel) and NLS-GFP (middle panel). Central slice from reconstructed ODT tomograms showing RI distributions (right panel). Bar on

the right shows the RI range. **e** Nuclear volume ($V_n$, gray) and dry mass ($M_n$, yellow) increase during nuclear assembly. Symbols show mean values. **f** Nuclei assembled in *Xenopus* extracts are import competent. Quantification of nuclear NLS-GFP fluorescence (dark green) during nuclear assembly. Symbols show mean values. **g** Nuclear mass density ($\rho_n$) reduces during assembly. As nuclei assemble, the mass density quickly drops within 5 min. Between 5 and 15 min, the nuclear mass density reduces below that of the cytoplasm and remains below cytoplasmic density up to 60 min. Black dashed line shows the mass density of the cytoplasm ($\rho_{cyto}$) and symbols show mean values. All scale bars: 5 μm. Black lines and bars in all graphs represent the mean ± SEM. Dashed lines in (**e**–**g**) are used to guide readers along the trend followed by the experimental data. Source data are provided as a Source Data file. Sample size and replicate information available in the Statistics and Reproducibility section.

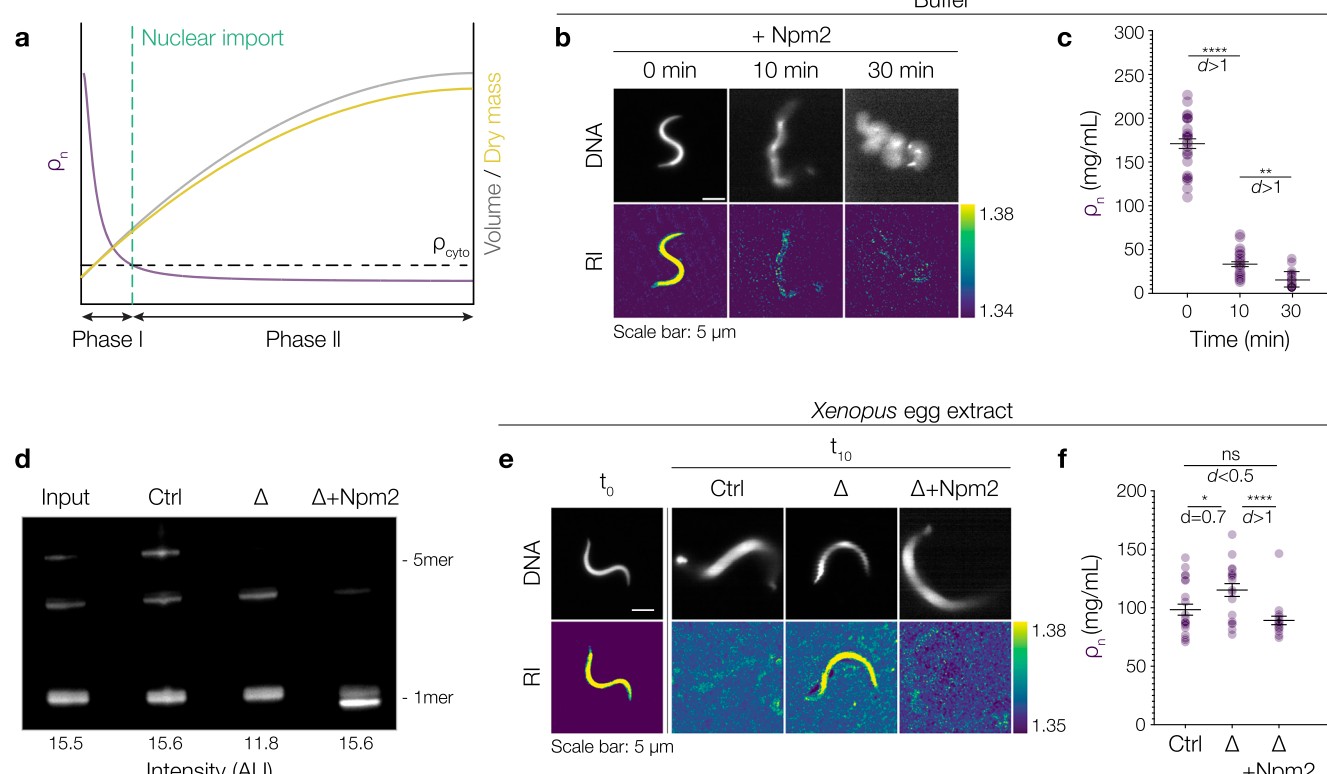

**Fig. 2 | Nucleoplasmin is both necessary and sufficient to reduce nuclear density to that of the surrounding solvent. a** Observed changes in nuclear density ($\rho_n$, purple), volume (gray), and dry mass (yellow) during nuclear assembly. Final nuclear density is lower than that of the cytoplasm ($\rho_{cyto}$, black dashed line). **b** Addition of Nucleoplasmin (Npm2) to sperm nuclei reduces RI and increases volume. Top panel shows DNA. Lower panel shows the RI images. Bar on the right shows the RI range. **c** Chromatin density ($\rho_n$, purple) reduces upon Npm2 addition. Circles represent individual data points. **d** Representative Npm2-immunoblot of *Xenopus* egg extract (input), control-depleted (Ctrl), Npm2-depleted (Δ), Δ with Npm2 addition (Δ+Npm2). Quantification of total intensity of all bands in the blot.

**e** Npm2 is both necessary and sufficient to reduce the density of sperm chromatin. Representative fluorescence (top panel) and RI images (bottom panel) of assembling nuclei from control-depleted, Δ and Δ+Npm2 extracts. **f** Chromatin density can be rescued by supplementing Npm2 after immunodepletion. Chromatin density ($\rho_n$, purple) 10 min after the start of nuclear assembly. Chromatin density remains high in Npm2-depleted extracts but recovers upon Npm2 addition. All scale bars: 5 μm. Black lines and bars in all graphs represent the mean ± SEM. Source data are provided as a Source Data file. Sample size and replicate information available in the Statistics and Reproducibility section.

deficient nuclei were significantly smaller when compared to control nuclei (Fig. 3c) consistent with previous reports[41,42]. Importantly, the density of the import-deficient nuclei ($\rho_{n\_60min} = 101.1 \pm 1.6$ mg/mL) did not reduce below the density of the cytoplasm implying that nuclear import is essential to further reduce nuclear density (Fig. 3a, d, e; Supplementary Notes Fig. 4). Consistently, when we simultaneously inhibited chromatin decondensation and nuclear pore formation, nuclei retained a density as high as that of condensed sperm chromatin (Supplementary Fig. 3a–d). Taken together, we have established that (1) Npm2-dependent chromatin decondensation quickly equalizes

nuclear density to that of the surrounding cytoplasm and that (2) nuclear import is essential to further decrease nuclear density below that of the cytoplasm. The question then arises how active import of macromolecules into the nucleus can establish a reduced nuclear density?

## Osmotically active solute macromolecules and chromatin determine nuclear volume and density

Nuclear import results in a reduction of nuclear density. This, at first glance, seems counterintuitive. How can the addition of molecules to

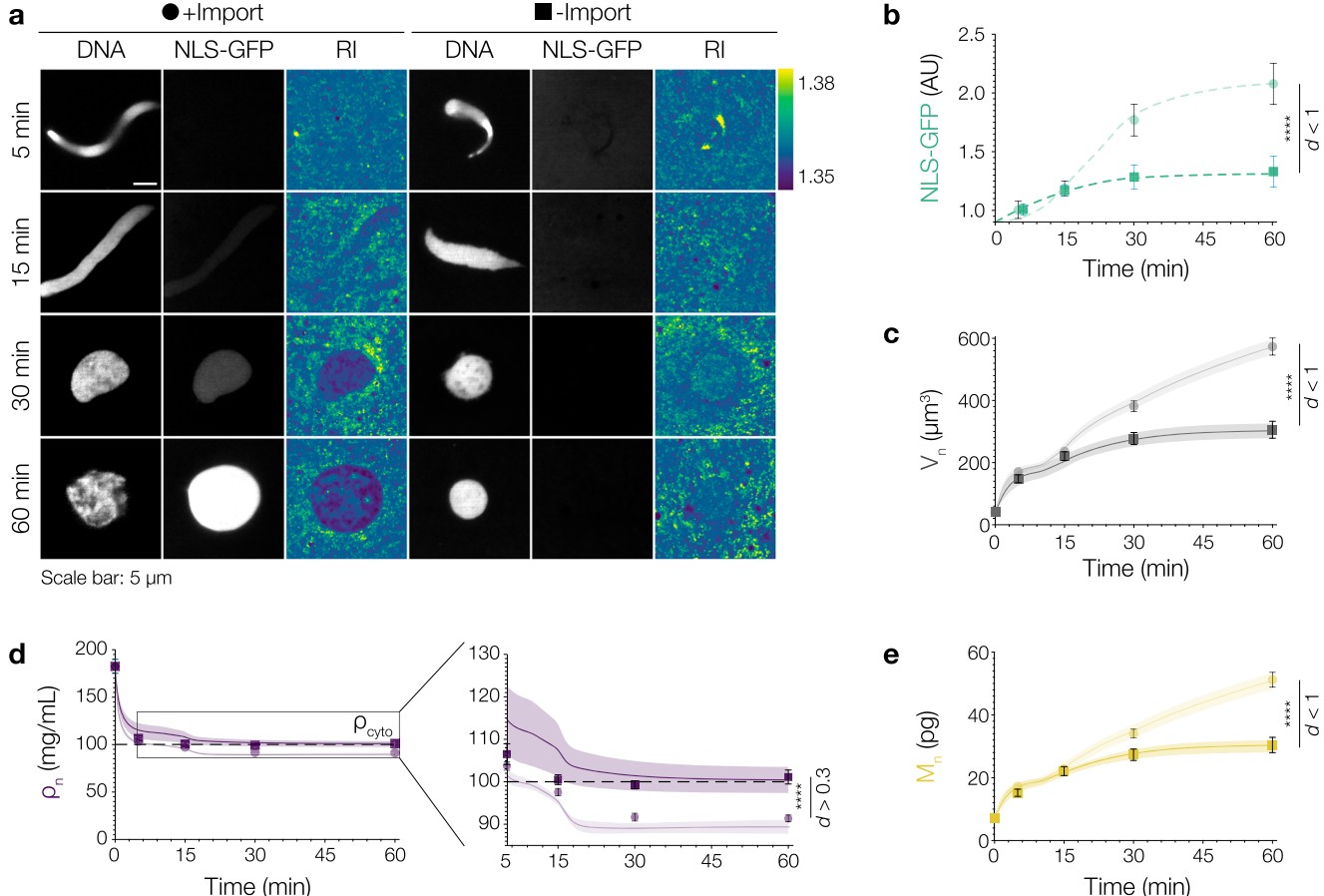

**Fig. 3 | Nuclear import is required to reduce nuclear density below that of the cytoplasm. a** Nuclei assembled in *Xenopus* egg extract in the absence (circles) or presence (squares) of import inhibitors (Ivermectin and Pitstop-2). Fluorescence images of DNA (left panel), NLS-GFP (middle panel) and RI images (right panel). Scale bar: 5 μm. Bar on the right shows the RI range. **b** Inhibitors successfully reduce nuclear import. Quantification of NLS-GFP import. Symbols represent the mean values. In (**b**−**e**), circles and lighter colors show the values for the +Import nuclei while squares and darker colors show the values for −Import nuclei. **c** Nuclear volume ($V_n$, gray) is reduced in import-deficient nuclei. At 60 min, there is a significant difference between the volumes of import-deficient and control nuclei.

Bold lines and shaded areas in (**c**−**e**) represent mean ± SEM from theoretical simulations (Supplementary Notes Fig. 4). **d** Nuclear density ($\rho_n$, purple) remains high in import-deficient nuclei. Import-deficient nuclei have a density, $\rho_n$, similar to that of the cytoplasm ($\rho_{cyto}$, black line). Inset shows a zoomed-in view of $\rho_n$ between 5 and 60 min. At 60 min, import-deficient nuclei still have a density similar to that of the cytoplasm. **e** Nuclear dry mass ($M_n$, yellow) is reduced in import-deficient nuclei. At 60 min, import deficient-nuclei have a lower dry mass than import-competent nuclei. Black lines and bars in all graphs represent the mean ± SEM. Source data are provided as a Source Data file. Sample size and replicate information available in the Statistics and Reproducibility section.

the nucleus make it less dense? It has been proposed that the import of nuclear proteins generates an osmotic pressure across the nuclear envelope that inflates the nucleus through water influx[43–45]. This has led to the distinct prediction that at steady-state the concentration of proteins in the nucleus and the cytoplasm should be equal[46–48]. To calculate the osmotic pressure exerted by nuclear and cytoplasmic proteins, we need to first realize that the osmotically active units are protein complexes (not monomeric proteins). To make this tangible, consider a ribosome: assembling 82 proteins and four RNAs into a single macromolecular complex reduces their osmotic pressure by a factor of 86[44]. Recently, proteome-wide analysis in *Xenopus* oocytes revealed the average mass of nuclear protein complexes to be 131.1 ± 1.4 kDa and 155.8 ± 1.5 kDa for cytoplasmic protein complexes[49]. Together with our density measurements, we estimated the concentration of protein complexes in the nucleoplasm $n_n$ and in the cytoplasm $n_c$ to be 0.51 ± 0.02 mM and 0.65 ± 0.02 mM, respectively (Fig. 4a). Accounting for protein complexes significantly lowered the number estimate with consequences for colloid osmotic pressures (Supplementary Notes Fig. 2 and Table 2). Based on the above concentrations, we then calculated the osmotic pressure exerted by nuclear and cytoplasmic proteins. The osmotic pressure exerted by nuclear proteins alone, however, was not sufficient to

draw enough water into the nucleus to explain the observed final nuclear volume and density. What are we missing? The most evident nuclear macromolecule that has the potential to exert additional pressure is chromatin[46,47,50]. To experimentally assess the contribution of chromatin, we assembled nuclei around either tetraploid *X. laevis* (genome size: 3.1 × 10⁹ bp) or diploid *X. tropicalis* (genome size: 1.7 × 10⁹ bp) sperm in an identical cytoplasm (Fig. 4b). As reported before[41,51], the *X. tropicalis* nuclei were smaller than *X. laevis* nuclei but importantly they had the same nuclear density (Fig. 4c–e, Supplementary Fig. 4a). As the extract system is transcriptionally and translationally inactive and the total number of nuclear pore complexes is similar in *X. laevis* and *X. tropicalis* nuclei[41], we propose that chromatin has a non-negligible effect on the pressure balance and thus nuclear size. In fact, chromatin can contribute to the pressure balance in two ways: (1) by exerting an outward pressure when the nuclear envelope confines its thermodynamically favored volume[50], and (2) as a regulator of nuclear import via the RanGTP gradient[52]. Indeed, when we included chromatin pressure to the pressure balance (Supplementary Eq. (5)), nuclear volume predictions quantitatively matched the measured experimental values (Fig. 4c, d). Thus, we estimated chromatin pressure to contribute to about 20% of the final size of *Xenopus* nuclei.

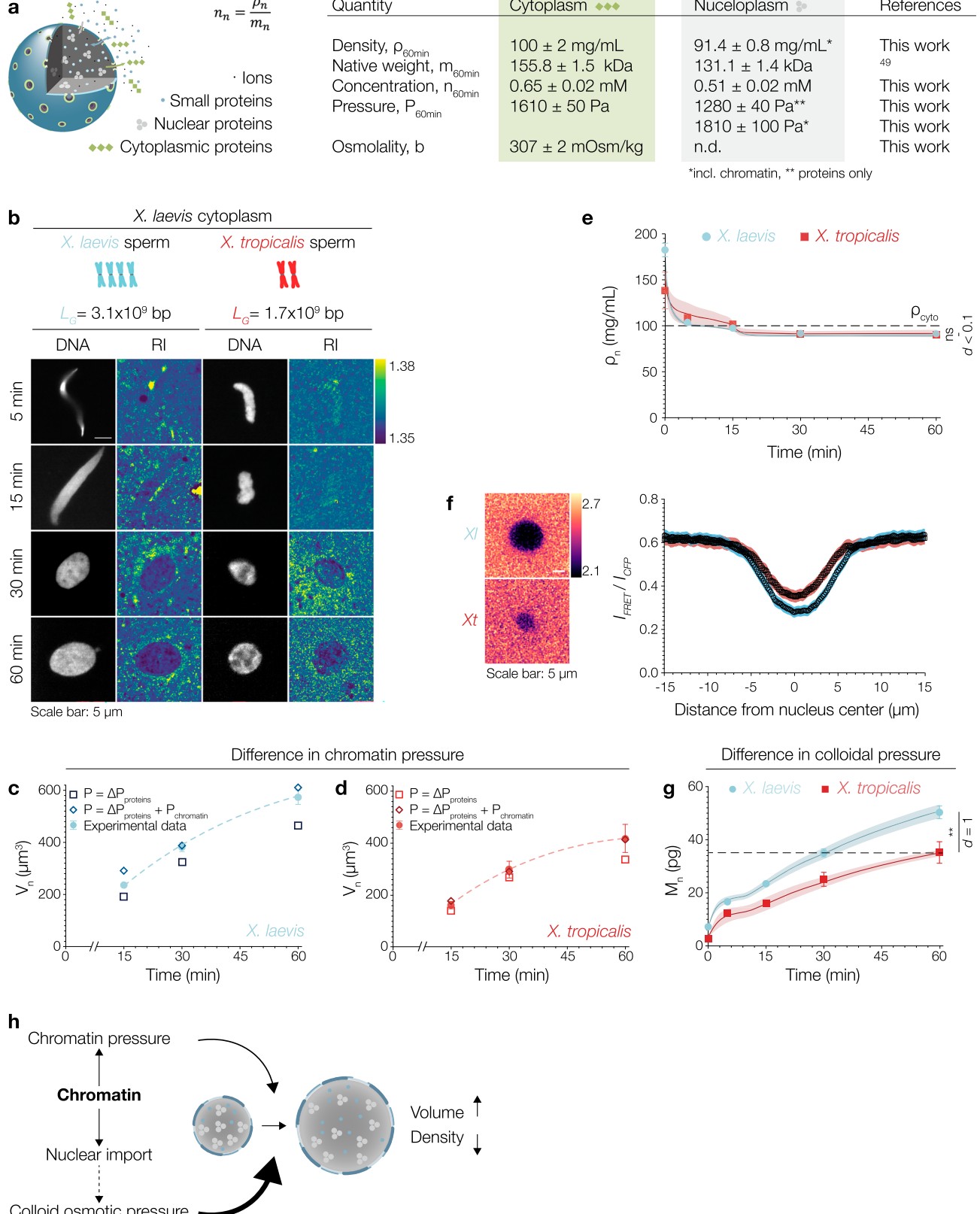

**a**

$$n_n = \frac{\rho_n}{m_n}$$

· Ions
· Small proteins
· Nuclear proteins
◆◆◆ Cytoplasmic proteins

| Quantity | Cytoplasm ◆◆◆ | Nucleoplasm ∴ | References |
|---|---|---|---|
| Density, $\rho_{60min}$ | 100 ± 2 mg/mL | 91.4 ± 0.8 mg/mL* | This work |
| Native weight, $m_{60min}$ | 155.8 ± 1.5 kDa | 131.1 ± 1.4 kDa | [49] |
| Concentration, $n_{60min}$ | 0.65 ± 0.02 mM | 0.51 ± 0.02 mM | This work |
| Pressure, $P_{60min}$ | 1610 ± 50 Pa | 1280 ± 40 Pa** | This work |
| | | 1810 ± 100 Pa* | This work |
| Osmolality, b | 307 ± 2 mOsm/kg | n.d. | This work |

*incl. chromatin, ** proteins only

**b** *X. laevis* cytoplasm

*X. laevis* sperm    *X. tropicalis* sperm

$L_G$ = 3.1x10⁹ bp    $L_G$ = 1.7x10⁹ bp

DNA    RI    DNA    RI

5 min / 15 min / 30 min / 60 min

Scale bar: 5 µm

**c** Difference in chromatin pressure

□ P = ΔP_proteins
◇ P = ΔP_proteins + P_chromatin
● Experimental data

$V_n$ (µm³) vs Time (min)    *X. laevis*

**d**

□ P = ΔP_proteins
◇ P = ΔP_proteins + P_chromatin
● Experimental data

$V_n$ (µm³) vs Time (min)    *X. tropicalis*

**e** ● *X. laevis*  ■ *X. tropicalis*

$\rho_n$ (mg/mL) vs Time (min)    $\rho_{cyto}$    ns $d < 0.1$

**f** Xl / Xt    Scale bar: 5 µm

$I_{FRET}/I_{CFP}$ vs Distance from nucleus center (µm)

**g** Difference in colloidal pressure

● *X. laevis*  ■ *X. tropicalis*

$M_n$ (pg) vs Time (min)    ** $d = 1$

**h**

Chromatin pressure
**Chromatin**
Nuclear import
Colloid osmotic pressure

Volume ↑
Density ↓

Next, to quantify the effect of chromatin content on nuclear import and final mass, we measured the RanGTP gradient, NLS-GFP accumulation, and nuclear dry mass. Consistent with previous reports[53], we observed a significant enrichment of RanGTP in the nucleoplasm. *X. laevis* nuclei showed higher RanGTP levels than *X. tropicalis* nuclei (Fig. 4f). Corroborating this, the accumulation of NLS-GFP, as a proxy for nuclear import efficiency, was also higher for *X. laevis* nuclei (Supplementary Fig. 4b) leading to a 1.4-fold increase in nuclear dry mass when compared to *X. tropicalis* nuclei (Fig. 4g). Collectively our data imply that the nuclear population of imported macromolecules creates a colloid osmotic pressure, which − assisted by chromatin pressure −

**Fig. 4 | Osmotically active solute macromolecules determine nuclear density and volume. a** Nuclear transport results in the establishment of a specific nuclear proteome that is distinct from the cytoplasmic proteome. Small proteins (<40 kDa), ions, and small metabolites - while being major osmolytes - can freely diffuse through nuclear pore complexes, which allows them to reach chemical equilibrium. The main contributors to the pressure difference across the nuclear envelope are proteins that preferentially locate to either the cytoplasm or nucleoplasm. Quantities measured and calculated to derive pressure balance and predict nuclear volumes. Further details in Supplementary Notes and Table 2 therein. **b** Nuclear assembly around *X. laevis* sperm (blue) or *X. tropicalis* sperm (red) in *X. laevis* extract. Fluorescence DNA images and corresponding RI images. Color scale: RI range. **c** Volume of *X. laevis* nuclei and **d** *X. tropicalis* increases over time. Experimental data points (circles) and predicted values from pressure balance. Considering pressure by proteins (squares) in (**c**, **d**) alone results in underestimation but including chromatin pressure (diamonds) matches experimental

values. **e** Density ($\rho_n$) of both nuclei decreases over time. After 60 min, the densities are comparable. Dashed black line indicates cytoplasmic density ($\rho_{cyto}$). Bold lines and shaded areas in (**e**, **g**) represent mean ± SEM from theoretical simulations (Supplementary Notes Fig. 6). **f** RanGTP levels in *X. laevis* (top) and *X. tropicalis* (bottom) nuclei using a FRET probe[53]. Color scale: fluorescence intensity range. Line scan quantifies FRET ratio signal (Intensity$_{FRET}$/Intensity$_{CFP}$). **g** Nuclear dry mass ($M_n$) of both nuclei increases over time. Dashed black line shows the dry mass in *X. tropicalis* nuclei at 60 min is similar to *X. laevis* nuclei at 30 min. **h** Schematic demonstrating how chromatin plays a dual role in pressure balance by (i) exerting a chromatin pressure due to confinement and further (ii) contributing to the colloid osmotic pressure by affecting nuclear import which is required to establish a distinct nuclear proteome. All symbols: Mean ± SEM. All scale bars: 5 μm. Source data are provided as a Source Data file. Sample size and replicate information available in the Statistics and Reproducibility section.

inflates the nucleus to its observed final volume and resulting density (Fig. 4h).

Despite the long-established scaling relation of nuclear size with DNA content[54], recent studies moved the focus towards cytoplasmic factors setting nuclear size independent of nuclear DNA content[41,47,48]. As we show here, these two concepts are not mutually exclusive and can thus be linked in a unifying framework. To this end, we developed a mechanistic model based on general biophysical principles of pressure balance and kinetics of active transport[55,56] allowing us to describe the full dynamics of nuclear growth (Supplementary Fig. 4c, Supplementary Movie 1, Supplementary Eqs. (25) and (26)). Our model correctly predicts the effects of biochemical perturbations, which include inhibition of nuclear import and replication, osmotic challenges and changes in chromatin content (Supplementary Fig. 4d–h and 5a–c, Supplementary Movie 1, Supplementary Notes Figs. 4–8). Up to now, we provided a mechanistic understanding of how the nucleus sets and maintains its density. The question of what makes the cytoplasm dense still remains.

## Reversal of the NC density ratio during senescence and in vitro is linked to heavy osmotically inactive cytoplasmic components

The cytoplasm is polydisperse in nature and enriched in multiple components and organelles that contribute to its density[57]. How local molecular composition defines subcellular density remains underexplored. In this respect, the loss of subcellular components has been suggested to contribute to cytoplasmic dilution and cellular senescence[27,58]. Whether senescence-induced cytoplasmic dilution has consequences on subcellular density distributions is unknown (Supplementary Fig. 6a). To investigate this, we induced senescence in human retinal epithelial cells (hTERT RPE-1) using the CDK4/6 inhibitor Palbociclib and found cells to be arrested in G1 with increased cell volumes and β-Gal levels, and with decreased cellular protein concentrations (Supplementary Fig. 6b–f) as reported previously[27,58,59]. In senescent cells, both cytoplasmic and nuclear densities decreased. However, while cytoplasmic density reduced by two-thirds from 66 to 22 mg/mL, nucleoplasmic density only reduced by half from 56 to 28 mg/mL (Supplementary Fig. 6g, h). This led to an inverted NC density ratio with the nucleus appearing denser than the cytoplasm (Fig. 5a, b). Of note, density ratios recovered upon drug removal (Fig. 5a, b, Supplementary Fig. 6g, h). Interestingly, nucleoli of senescent cells stood out as they were enlarged (Fig. 5c), which we confirmed by measuring nucleolar volume and dry mass (Fig. 5d, e) consistent with previous reports[60,61]. While we found the local concentration of pre-ribosomal subunits to be enriched in nucleoli, ribosome purification profiles revealed a global reduction in mature subunits and ribosomal complexes in senescent cells (Fig. 5f–i, Supplementary Fig. 6i–l). This data supports the role of ribosomes in cytoplasmic crowding[62–64] and further proposes that cytoplasmic dilution is due to perturbed ribosomal subunit biogenesis and export during senescence[65,66].

Is the reduction in ribosome number sufficient to reduce cytoplasmic density or is it an indirect effect of reduced ribosome function? To test this, we assembled nuclei in *Xenopus* high-speed (HS) extract that lacks heavy components including ribosomes, which lowers extract density but does not change its osmolality (Fig. 5k, l). Importantly, nuclei assembled in this minimal cytoplasm were import- and replication competent (Supplementary Fig. 6m, n [67,68]) and had a nuclear dry mass identical to that of control nuclei (Fig. 5m, LS = 51 ± 2 pg and HS 48 ± 4 pg, respectively) which implies that they import the same total amount of proteins as control nuclei. As a consequence, these nuclei appeared denser in a diluted cytoplasm. Adding back purified ribosomes increased cytoplasmic density but only the combined addition of purified ribosomes and glycogen − a known cytoplasmic crowder[57] − fully restored cytoplasmic density and thus the NC density ratio (Fig. 5n, o). While in somatic cells ribosome concentration might primarily tune cytoplasmic density, the specific density of a maternal cytoplasm such as the *Xenopus* egg extract is set by ribosome and glycogen concentrations. Taken together, these results show that cytoplasmic density is set by heavy components, their relative composition might determine cell-type specific differences in density (Fig. 1b). Next, to substantiate our mechanistic understanding, we wished to obtain independent experimental support from an in vivo system, in which nuclear volumes change during development.

## During early development, the nuclear-to-cytoplasmic *density* ratio is robustly maintained while *volume* ratios change

Contrary to the NC *density* ratio, the well-known NC *volume* ratio has been studied extensively ever since first described by Richard Hertwig as early as 1903[31]. Although NC volume ratios are reported to be constant, they can change when cells change fate or size during development and differentiation[69,70]. What happens to the NC density ratio when NC volume ratios change? To study how nuclear and cytoplasmic densities respond to changing NC volume ratios, we imaged early divisions in *C. elegans* embryos via ODT (Fig. 6a, Supplementary Movie 2). As cell number increased, cell volume decreased, nuclear size scaled with cell volume (Fig. 6b, Supplementary Fig. 7a) and NC volume ratios changed (Fig. 6d) consistent with previous reports[71,72]. To iterate the role of chromatin, we in addition imaged tetraploid embryos (Fig. 6a, Supplementary Movie 2). Consistent with our results in *Xenopus* egg extracts, higher ploidy led to larger nuclei (Fig. 6c). Interestingly, not only nuclear size but also cell and embryo size scaled with ploidy spanning four hierarchical levels of biological organization (Supplementary Fig. 7c–f)[73]. The overall nuclear scaling behavior of diploid and tetraploid embryos, however, was comparable (Fig. 6d, e, Supplementary Fig. 7a, b). Interestingly, nuclear and cytoplasmic densities were constant over the course of the first four divisions in both worm strains (Fig. 6f–i) with the nucleus always being less dense than the cytoplasm. Thus, while the NC volume ratios increased, the NC density ratio was robustly maintained (Fig. 6j, k). Importantly, these

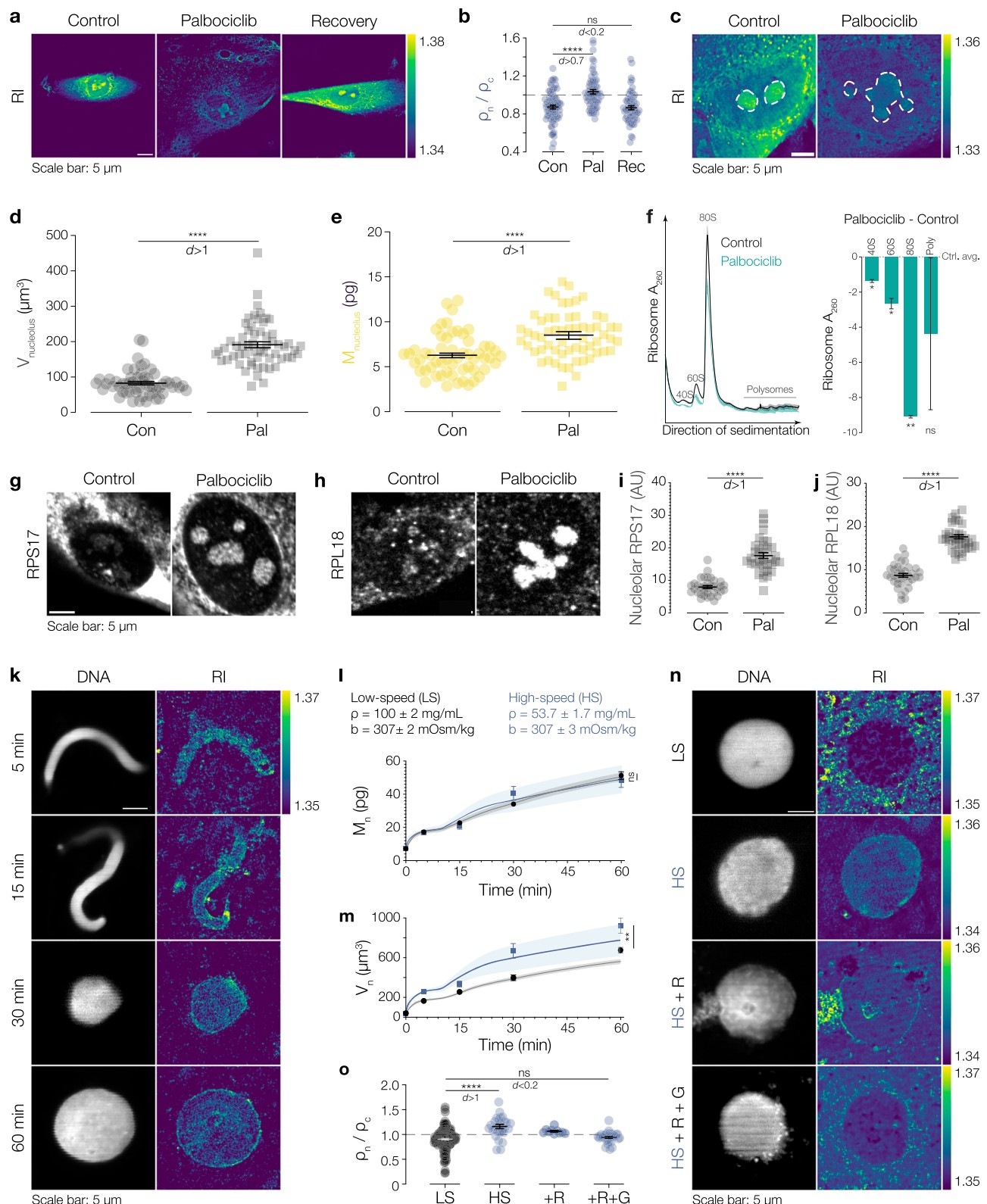

observations are fully consistent with our pressure balance model: absolute nuclear size decreases because the maternal cytoplasm of the one-cell embryo is partitioned into smaller cells at constant density (Fig. 6f, g). Simply, in smaller cells, the material available for nuclear growth is reduced. While the absolute dry mass and volume of nuclei decreases, the absolute dry mass and volume of the cytoplasm decreased even more rapidly (Supplementary Fig. 7g, h). Thus, to keep a constant NC density ratio and maintain the pressure balance across

the nuclear envelope, the relative nuclear size and mass have to increase (Fig. 6l). Accordingly, nuclei occupy a higher percentage of the cell volume while the density ratio is maintained. These observations imply that the NC volume ratio, in fact, is a consequence of cells maintaining a constant NC density ratio with broad implications for cellular physiology[27]. Taken together, our observations in *C. elegans* embryos showed that the NC density ratio is maintained in the lineage of an embryonic system. They confirm our data from the in vitro

**Fig. 5 | Reversal of the NC density ratio during senescence and in vitro is linked to heavy osmotically inactive cytoplasmic components. a** Representative RI images of control (Con), senescent (Palbociclib treated, Pal) and recovering (Rec) hTERT RPE-1 cells. **b** Quantification of NC density ratio. Symbols represent the average value of each cell. **c** Nucleoli appear larger during senescence. Representative RI images showing nucleoli (dashed white lines). **d** Nucleolar volume ($V_{nucleolus}$) and **e** dry mass ($M_{nucleolus}$) increase during senescence. Symbols represent the total dry mass or size of nucleoli of one cell. **f** Ribosome profiles indicate reduction in ribosomal subunits and mature ribosomes during senescence. Mean ± SD of Con (black) and Pal (green) profiles. Positions of 40S, 60S, 80S, and polysome peaks are annotated. Quantification of difference between Con and Pal, for the $A_{260}$ sum of ribosome peaks. **g, h** Immunofluorescence and **i, j** quantification shows increased accumulation of RPS17 (40S ribosomal protein S17) and RPL18 (60S ribosomal protein L18) pre-ribosomal subunits in nucleoli during senescence. **k** Nuclear assembly in high-speed (HS) extract. Left panel: DNA stained with Hoechst-33342. Right panel: RI images. **l** Nuclear dry mass in HS extracts and low-speed (LS) extracts is similar. In (**l, m**) black circles: LS nuclei and blue squares: HS nuclei. In (**l, m**) bold lines and shaded areas represent mean ± SEM from theoretical simulations. See also Supplementary Notes Fig. 7. **m** Quantification of nuclear volume during nuclear assembly in HS. **n** NC density ratios can be altered by modifying cytoplasmic complexity. Representative images of nuclei assembled in LS, HS or HS supplemented with purified ribosomes (HS + R) or both ribosomes & glycogen (HS + R + G). Left panel: DNA stained with Hoechst-33342. Right panel: RI image. **o** NC density ratios recover when adding back heavy components. Quantification of ratio for each condition. Each symbol represents the value from one nucleus. Black dashed line shows cytoplasmic density. Scale bars: 5 μm. Color scales: RI range. Black bars and lines indicate mean ± SEM. Source data are provided as a Source Data file. Sample size and replicate information available in the Statistics and Reproducibility section.

reconstitutions in *Xenopus* and substantiate our mechanistic view on nuclear formation and growth. They further strengthen our pressure balance model which quantitatively links ploidy, nuclear dry mass, density, and volume.

## Discussion

While it is well appreciated that the physical and chemical properties of the cytoplasm and the nucleoplasm have far-reaching consequences for cellular function, subcellular density distributions remain largely unexplored. This study provides a mechanistic and quantitative understanding of how nuclei establish and maintain a lower density than the surrounding cytoplasm. Active import of nuclear proteins − assisted by chromatin pressure − establishes a pressure difference across the nuclear envelope, which expands the nucleus to its final volume and resulting lower density. We show that a lower nuclear density with a constant NC *density* ratio is robustly maintained even during early development when NC *volume* ratios change. This implies that cells maintain a less crowded nucleus by adjusting their nuclear volume to cell volume. We find such a constant NC density ratio of approximately 0.8 in 10 cell types ranging from yeast to human, which suggests that this ratio is a fundamental property ubiquitous to living systems. Deviations from the conserved NC density ratio are found during senescence and indicative of pathophysiology.

In 1903, Richard Hertwig formulated the first quantitative hypothesis on organelle size[31]. Based on his work on sea urchin embryos and algae, he proposed the "Kern-Plasma-Relation" (NC volume ratio) to be a constant characteristic for a given cell type. Here, we identify a conserved ratio of NC density, which we propose to be the biophysical driver of the repeatedly observed "Kern-Plasma-Relation"[41,45−48,69−74]. Our experimental data together with the pressure balance model provide critical quantitative evidence for a mechanism which intrinsically links nuclear size to cell size by establishing a specific NC density ratio. Furthermore, we provide several important biophysical quantities, e.g. nuclear and cytoplasmic density, dry mass, protein complex concentrations, and colloid osmotic pressures, which in the future can help constrain and advance many conceptual and theoretical models of nuclear size control and cellular density[17,27,45−48].

In *Xenopus* egg extracts, our proposed model of pressure balance by localized nuclear proteins and chromatin sufficiently explains final nuclear size and density. It is, however, important to emphasize that using *Xenopus* egg extracts as a model system allows us to make simplifying assumptions. Cellular reality is likely more complex. Previous studies have characterized a multitude of cellular processes being involved in setting nuclear size. These include, among others, nucleocytoplasmic transport[41,75], gene expression and RNA processing[76], limiting cytoplasmic factors[41,77], mechanical coupling between the cytoskeleton, the nuclear envelope, and chromatin[47,78], and lipid homeostasis[79]. Thus, the proposed pressure balance will likely act in concert with these cellular processes and be constrained

by other mechanical elements in vivo[80]. Our model is qualitatively consistent with osmotic-based models of nuclear size scaling[19,46−48] including the "Pump-and-Leak" model[46]. Nonetheless, it is crucial to recognize that the system parameters can significantly vary across different biological systems, e.g. cytoplasmic and nuclear densities as shown in Fig. 1 and this variation also carries over to the pressure values (see Section 1.1.4 in Supplementary Note - Theory). Therefore, an accurate, system-specific accounting of the biophysical parameters will be an essential component in future modeling efforts together with developing more advanced models pertinent to the biological system in question.

In this study, we explain how the unfolding and replicating chromatin contributes to the pressure balance both directly as a confined polymer and indirectly as a regulator of nuclear import. We thus propose that chromatin pressure has a modest but non-negligible direct effect on nuclear size. In our model chromatin pressure originates from its confinement by the nuclear envelope. It has, however, been postulated that charged macromolecules including chromatin are surrounded by counterions to ensure electroneutrality[46,47,81]. Collectively, these counterions could exert a substantial osmotic pressure. While some argue that under crowded conditions, osmotic pressures generated by counterions are negligible[44], our experiments do not allow us to discriminate between the pressure of chromatin due to confinement and the osmotic pressure exerted by chromatin counterions.

Finally, the question remains: why would it be important to maintain a robust density ratio between the nucleus and the cytoplasm? For one, it could simply be a physical consequence of the relative distributions of protein identities in the nucleus and the cytoplasm. We found absolute densities to be profoundly distinct (Fig. 1b), this could imply that evolution shaped the cell-type specific proteome to optimize the trade-off between total protein concentration and colloid osmotic pressure[44,82]. Potentially though, a constant NC density ratio evolved to serve a function. One conceptual idea is that a constant NC density ratio is essential for a homeostatic coupling of transcription and translation[26,58,83,84]. Indeed, it has recently been proposed that the maintenance of a homeostatic cell density is due to a scaling relation between amino acids, which are major osmolytes in the cell, and proteins, which contribute most of a cell's dry mass[46]. This is consistent with our and previous observations of a diluted cytoplasm in senescent cells, which fail to scale nucleic acid and protein biosynthesis with cell volume[27]. Understanding how cell type-specific density and density distributions are set by balancing transcription, ribosome biogenesis, protein synthesis, and transport rates will remain an exciting avenue for future research.

## Methods

### *Xenopus spec*

The *Xenopus* frogs (adult females or males) used in this study are part of the *Xenopus* colony maintained at the animal husbandry of the

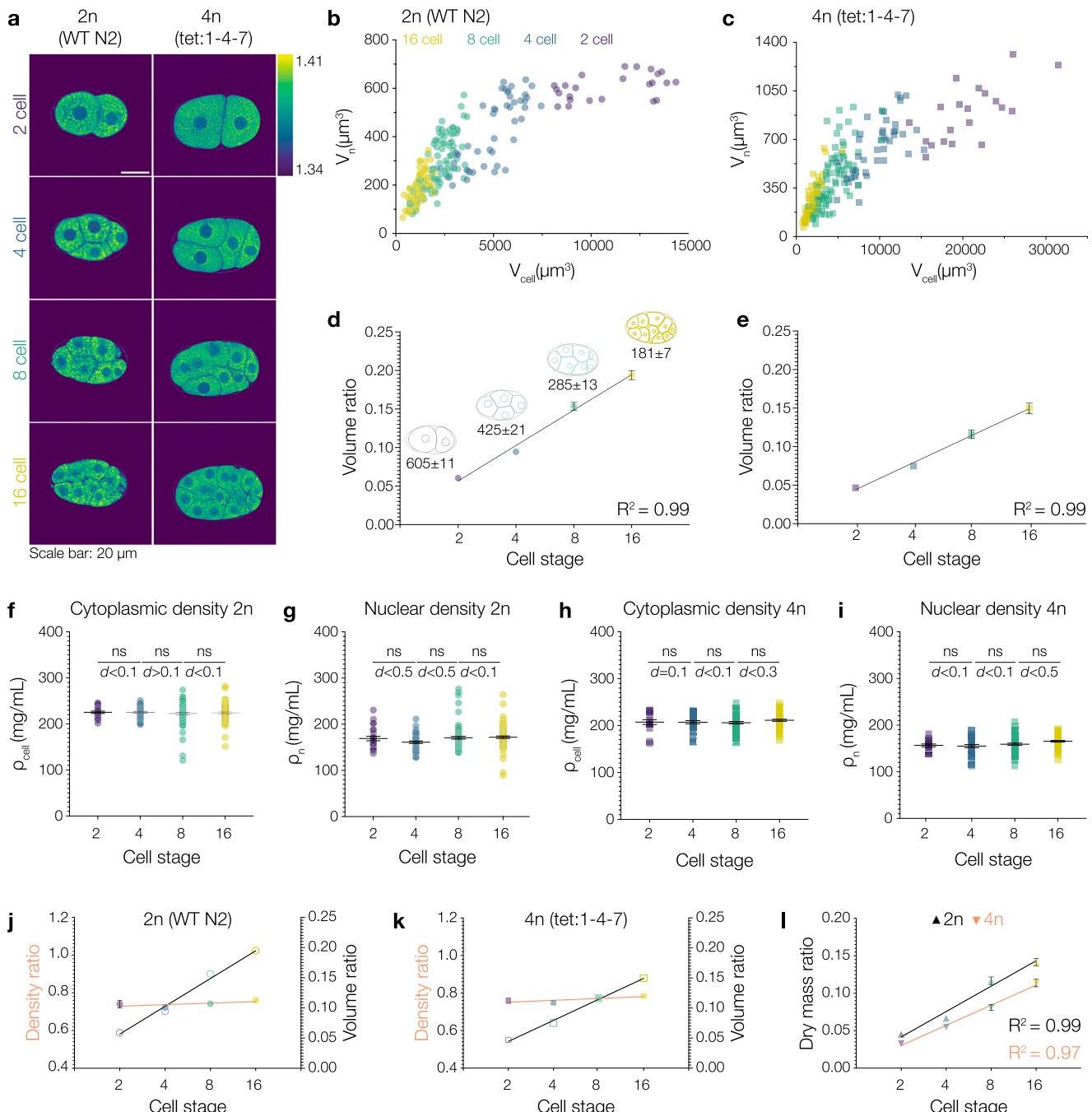

**Fig. 6 | During early development, the nuclear-to-cytoplasmic density ratio is robustly maintained even when volume ratios change. a** ODT can be used to study density distributions during early development in *C. elegans* embryos. Representative ODT images of *C. elegans* wildtype (left panel) and tetraploid (right panel) embryos. Scale bar: 20 μm. Color bar on the right shows RI distribution. **b** Nuclear volume ($V_n$) scales with cell volume in wildtype and **c** tetraploid embryos. Purple, blue, green, and yellow symbols represent values from 2 cell stage, 4 cell stage, 8 cell stage and 16 cell stage cells, respectively. **d** NC volume ratios change during development in wildtype and **e** tetraploid embryos. Average nuclear volumes (mean ± SEM) for each stage. Linear fit of volume ratios. **f** Cytoplasmic and **g** nuclear density is conserved through development in wildtype and tetraploid (**h**, **i**) embryos. Each circle shows the value from a single cell. The NC density ratios is maintained even when NC volume ratios change during development in **j** diploid and **k** tetraploid embryos. Orange lines indicate linear fits of the density ratios. Black lines indicate linear fit of volume ratios. **l** The nuclear dry mass ratio increases during development in both wildtype (triangles) and tetraploid (inverted triangles) embryos. Bold black and orange lines indicate a linear fit of the data for the wildtype and tetraploid, respectively. Black lines and bars in all graphs represent the mean ± SEM. Source data are provided as a Source Data file. Sample size and replicate information available in the Statistics and Reproducibility section.

Humboldt-Universität zu Berlin and were obtained from NASCO (Fort Atkinson, WI). *Xenopus* frogs were maintained in a recirculating tank system with regularly monitored temperature and water quality (pH, conductivity, and nitrate/nitrite levels) at 18–20 °C (*X. laevis*) or 24–26 °C (*X. tropicalis*). All experimental protocols involving frogs were performed in accordance with national regulatory standards and ethical rules and reviewed and approved by the LaGeSo under Reg.-Nr. Reg 0113/20. XL-177 cells were maintained in DMEM (41966-029; Gibco) supplemented with 10% FBS and antibiotic–antimycotic (Invitrogen) in an incubator maintained at 27 °C with 5% $CO_2$ and passaged routinely. For imaging, cells were transferred onto imaging dishes (81156, μ-Dish 35 mm, ibidi).

## C. elegans

Wildtype N2 strain worms (diploid) were picked and grown on NGM plates at 20 °C. Single worms from strain SP346 (provided by the CGC, which is funded by NIH Office of Research Infrastructure Programs, P40 OD010440) were picked and grown for several generations at 16 °C on NGM/OP50 by selecting phenotypically long worms. Lines with consistent long phenotype and good fertility were selected. Oocyte chromosome numbers were counted to confirm tetraploidy of all lines. Line 1-4-7 was selected for all further experiments.

## S. cerevisiae

A haploid strain originally derived from the S288c strain (Genotype: $MAT\alpha$ $SUC2$ $mal$ $mel$ $gal2$ $CUP1$) was used for imaging. S. cerevisiae were recovered from −80 °C glycerol stocks onto a 2% YPD plate. Colonies were inoculated into 5 mL SD media (0.17% Yeast Nitrogen Base Without Amino Acids and Ammonium Sulfate, 0.5% Ammonium Sulfate, 2% Glucose) and incubated at 30 °C overnight with agitation. For imaging, cells were attached to imaging dishes (81156, μ-Dish 35 mm, ibidi) containing fresh SD media using Concanavalin A (C2010, Sigma; 1 mg/mL).

## S. pombe

S. pombe strain AEP1 (Genotype: FY7385; h- leu1-32 ura4-D18 his3-D3) was grown in full medium (YES) at 30 °C. For imaging, cells were transferred to imaging dishes (81156, μ-Dish 35 mm, ibidi) that were coated with Poly-L-Lysine (0.1% w/v, P6407, Sigma).

## C. reinhardtii

Motile Chlamydomonas reinhardtii wild-type strain CC125 were maintained in standard Tris-acetate phosphate (TAP) medium under continuous shaking at 110 rpm at 22 °C. For imaging, cells were transferred to imaging dishes (81156, μ-Dish 35 mm, ibidi) that were coated with Poly-L-Lysine (0.1% w/v, P6407, Sigma).

## D. melanogaster

Drosophila S2R+ cells were maintained in Schneider's medium (11720-034, Gibco) at 23 °C and 33% relative humidity. Cells were detached from T-75 tissue culture flasks by thoroughly pipetting the medium within the flask. For imaging, cells were allowed to attach to imaging dishes (81156, μ-Dish 35 mm, ibidi) that were pre-coated with Poly-L-Lysine (0.1% w/v, P6407, Sigma).

## D. rerio

Fertilized embryos were obtained from AB strain zebrafish. Embryos were allowed to grow for 4 hours post fertilization in E3 medium. Embryos at the dome stage 4–5 were collected and dissociated by gently pipetting up and down in deyolking buffer (55 mM NaCl, 1.8 mM KCl, 1.25 mM NaHCO₃ in HBSS, Life Technologies). For imaging, dissociated embryonic cells were transferred to imaging dishes (81156, μ-Dish 35 mm, ibidi).

## M. musculus

R1/E mouse embryonic stem cells were cultured as in ref. 85. Briefly, cells were maintained in DMEM media (41966-029; Gibco) supplemented with 16% FBS (Gibco), antibiotic–antimycotic (Invitrogen), nonessential amino acids (Gibco), β-mercaptoethanol (Gibco), and recombinant mouse leukemia inhibitory factor (ESGRO). Cells were passaged every 48 h and seeded at a density of 35,000 cells/cm² onto gelatin-coated dishes. For imaging, cells were transferred to imaging dishes (81156, μ-Dish 35 mm, ibidi) that had been coated with Laminin-511 (2.5 μg/mL, LN511, BioLamina)

## H. sapiens

HEK293 cells were cultured in DMEM (41966-029; Gibco) supplemented with 10% FBS and antibiotic–antimycotic (Invitrogen) in an incubator maintained at 37 °C with 5% $CO_2$ and passaged routinely. For imaging, cells were transferred onto imaging dishes (81156, μ-Dish 35 mm, ibidi).

hTERT RPE-1/RPE-1 FUCCI cell lines were cultured in DMEM F-12 (21041033; Gibco) supplemented with 10% FBS and antibiotic–antimycotic (Invitrogen) in an incubator maintained at 37 °C with 5% $CO_2$ and passaged routinely. For senescence induction the CDK4/6 inhibitor Palbociclib (final concentration: 4 μM) was added to the culture media. Cells were maintained in an inhibitor containing medium for 48 h. For recovery experiments, cells were washed 3 times with inhibitor-free medium and maintained for an additional 24 h. Care was taken to ensure cells did not become confluent during the senescence induction and recovery experiments. For imaging, cells were transferred onto imaging dishes (81156, μ-Dish 35 mm, ibidi). For total protein staining of live cells the CellTrace™ Yellow dye (C34567, Thermo Fisher, final concentration: 5 μM) was utilized as per the manufacturer's instructions.

## Immunofluorescence for detecting ribosomal subunits

Immunofluorescence was performed using primary antibodies for proteins RPS17 and RPL18 (1:200 Sigma HPA055060-100UL and HPA046572-100UL, respectively). Cells exposed to different conditions were seeded onto 24 well plates (82426, μ-Plate 24 well, ibidi). Cells were incubated with blocking solution (5% goat serum, 0.5% (v/v) Triton X-100, PBS) for 1 h at room temperature. Cells were subsequently incubated with primary antibody in blocking buffer for 16–20 h at 4 °C, washed in PBS three times for 30 min and incubated with secondary antibody diluted in the blocking buffer containing Hoechst-33342 for up to 4 h at room temperature. Cells were either imaged immediately or stored in PBS.

## Cell lysis and ribosome profile analysis

For each condition $6 \times 10^6$ cells (counted using Countess™ Cell Counter, Thermo Fisher) were collected in falcon tubes and snap frozen for further analysis. Frozen cell pellets were resuspended in 150 μL lysis buffer (20 mM HEPES pH 7.5, 100 mM KCl, 10 mM MgCl₂, 5 mM DTT (Dithiothreitol), 0.04 mM Spermine, 0.5 mM Spermidine, 1Tab Protease Inhibitor cOmplete EDTA-free (Roche, 05056489001) /100 mL, 0.5% NP-40) and incubated on ice for 30 min. The suspension was transferred to 1.5 mL reaction tubes and centrifuged for 2 min at $18,500 \times g$ and 4 °C. The concentration of the supernatant was determined by Nanodrop (control_1: 12.63 A₂₆₀/mL, Pal_1: 5.60 A₂₆₀/mL, control_2: 10.80 A₂₆₀/mL, Pal_2: 7.50 A₂₆₀/mL). Sucrose gradients (5–45% sucrose in lysis buffer without NP40) were generated in SW40 tubes using a Biocomp gradient master™. 200 μL of each sample were layered on the sucrose gradients and centrifuged in a SW40 rotor (Beckman Coulter) for 5 h at $80,000 \times g$ and 4 °C. A₂₆₀ profiles were collected using a Biocomp Piston Gradient Fractionator™ equipped with a Triax™ flow cell.

## Imaging and image analysis

**Correlative optical diffraction tomography and confocal fluorescence microscopy.** Refractive index (RI) tomograms were obtained using a custom-built optical diffraction tomography (ODT) setup based on Mach-Zehnder interferometry. The setup was built onto a commercial inverted microscope stand (IX81, Olympus Life Science), which increased usability and allowed us to use separate light paths for ODT, epifluorescence, and confocal fluorescence microscopy (Supplementary Fig. 1a). The detailed description of a similar ODT setup can be found in refs. 23,32. Briefly, a coherent laser beam (λ = 532 nm) was split into two using a single-mode fiber optic coupler. One beam was used as a reference beam and the other was used to illuminate the sample using a tube lens (f = 175 mm) and a 60× water-dipping objective lens (LUMPLFLN60XW, NA 1.0, Olympus Life Science). To reconstruct 3D RI tomograms, the sample was illuminated from 150 different

incident angles by a dual-axis galvanometer mirror (GVS012/M, Thorlabs Inc.) covering the azimuthal angular range from −48° to 48°. The scattered light from the sample was collected by a high numerical aperture 100× objective lens (oil immersion, UPlanFl, NA 1.3, Olympus Life Science) and interfered with the reference beam at the image plane, which generated spatially modulated holograms. The holograms were recorded by a CCD camera (FL3-U3-13Y3M-C, FLIR Systems, Inc.). The detailed principle for tomogram reconstruction can be found in refs. 86,87. Briefly, from the spatially modulated holograms, the complex optical fields of light diffracted by the sample were retrieved by applying a field retrieval method based on Fourier transformation[88]. The 3D RI tomogram of the sample was reconstructed by mapping 2D Fourier spectra of the retrieved complex optical fields onto the surface of Ewald spheres corresponding to the spatial frequency of the incident angles based on Fourier diffraction theorem[86–90]. The Gerchberg-Papoulis (GP) constraint is applied to fill the missing cone artifact.

Confocal fluorescence microscopy was performed using a Rescan Confocal Microscope (ref. 91, RCM1 from confocal.nl) which was installed on the same microscope frame. Fluorescence images or image stacks (with a step size of 1 μm) were acquired using the same high numerical aperture objective lens (UPlanFl, NA 1.3, Olympus Life Science) and recorded on to a CMOS camera (CM3-U3-50S5M-CS, FLIR Systems, Inc. or FL-AB-20-BW Tucsen Photonics Co., Ltd.). The RCM1 and microscope frame was controlled using a custom Micromanager script (provided by confocal.nl). For immunofluorescence, a spinning disk confocal fluorescence system was utilized. Imaging was performed on a Nikon Widefield Ti2 fluorescence microscope equipped with an Andor Revolution SD system, emCCD camera (iXon3 DU-888 Ultra) and a 60x oil immersion objective lens (CFI Plan Apochromat Lambda D 60X Oil). The system was controlled using the Nikon Elements AR software.

**Calculating mass densities from refractive index values.** 3D RI tomograms were reconstructed using a custom Matlab script[23,32]. The 3D tomogram was computed based on Fourier diffraction theorem[90] from multiple 2D quantitative phase images that were acquired by illuminating the sample at various oblique angles. Upon tomogram reconstruction each voxel within the 3D RI tomogram has an assigned RI value. The absolute RI values for the surrounding medium ($n_i$, where i = extract or medium) were independently measured using an Abbe refractometer (Arcarda ABBE-2WAJ). Since the RI value in most biological samples is linearly proportional to the mass density of material with the proportionality coefficient, or RI increment, $\alpha$[92], mass densities within each voxel were calculated using the relationship $n_{sample} = n_i + \alpha\rho_{sample}$, where $\alpha$ is the RI increment for proteins/nucleic acids (0.19 mg/mL[93]) and $\rho_{sample}$ is the mass density within the sample. For control extract samples, the mass density of $n_i$ (RI of the surrounding cytoplasm) was measured to be 100 mg/mL (based on protein concentration measurements using the Bradford assay). During perturbation experiments, the mass density value of $n_i$ was determined based on the measured protein concentration of the sample (such as for HS extracts or during osmotic shock). For measurements in buffer and cell culture medium, the RI of the medium ($n_i$) was measured using an ABBE refractometer and used as a background reference.

**Image analysis and quantification.** Nuclear/nucleolar/cytoplasmic regions of interest (ROI) determination and nuclear volume segmentation was performed using FIJI[94]. Within cells from different species, nuclear and cytoplasmic mass densities were measured using square ROI (0.5 μm length for *S. cerivasiae* and *S. pombe*, 1 μm length for *C. reinhardtii*, 5 μm length for *C. elegans* and *D. rerio*, 2 μm length for *D. melanogaster* and *X. laevis*, and 2.5 μm length for *M. musculus* and *H. sapiens*). For each cell, five arbitrary and unbiased selections were

made for each nuclear and cytoplasmic ROI. For nuclei containing nucleoli, nucleolar regions were excluded from the analysis of nucleoplasmic mass density. To calculate nucleolar volumes and density in senescent cells, RI images were segmented using the 3D volume manager plugin (part of the Bio Image Analysis Toolbox from the SCF, MPI-CBG; https://sites.imagej.net/SCF-MPI-CBG/). To calculate cell volume of mammalian cells Otsu thresholding was used to segment the cell from the surrounding medium followed by integrating the pixel volumes in the segmented region. For the immunofluorescence images, total fluorescence intensity in each compartment (cell, nucleus or nucleolus) was calculated after segmenting the image either using Otsu thresholding or the 3D volume manager plugin. Concentrations of ribosomal subunits were calculated for each cell by dividing the total fluorescence intensity (in arbitrary units) by its volume. Nuclear volumes were calculated using either fluorescent images of DNA stained with Hoechst-33342 or segmenting using the 3D volume manager.

For *Xenopus* egg extracts, nuclei were segmented from the 3D tomograms either manually or using the 3D volume manager plugin. To calculate nuclear volumes, the voxels within the segmented regions were integrated. To calculate nuclear dry mass, the mass densities within each segmented volume were integrated. To calculate the average mass density within each segmented nucleus, the total dry mass was divided by the total volume. To quantify fluorescence intensities (Hoechst-33342/NLS-GFP) nuclei within images were segmented either manually or using Otsu thresholding. To calculate fluorescence intensities, the gray values within the segmented regions were integrated. Next, the background intensities were calculated for regions outside nuclei with dimensions similar to the segmented nuclear regions. Finally, the total fluorescence intensity within the sample was divided by the total background intensity and these values were reported in the figures.

**C. *elegans* embryo ODT imaging and analysis.** Custom-built ODT setups were used to image wildtype and tetraploid embryos. Embryos were isolated from healthy adults and placed onto coverslips (631-1573P, VWR International). A drop of M9 buffer (~10 μL) containing 20 μm polystyrene beads was placed on the coverslip along with the embryos. A second smaller coverslip (631-0124, VWR International) was gently placed on top of the embryo to create a confined volume for imaging which reduced scattering. The beads acted as spacers and prevented squashing of the developing embryos.

For *C. elegans* embryos in addition to the regular field retrieval and tomogram reconstruction code, an axial drift during the time lapse imaging was corrected by identifying the center of mass of the reconstructed RI tomograms. Further, since the embryos were placed between 20 μm beads, axial constraints were implemented within the code to reduce scattering and improve the quality of the reconstructed tomograms. Embryos and nuclei were segmented using the 3D volume manager plugin and the segmented masks were used for calculating volumes. The average mass density within the segmented volumes were reported and total dry mass was calculated by integrating the mass density information over the 3D volume. The NC volume ratio was calculated by dividing the nuclear volume by the cytoplasmic volume (cell volume − nuclear volume). The NC density ratio was calculated by dividing the average mass density of the nucleus by the average mass density of the cytoplasm. The dry mass ratio was calculated by dividing the total nuclear dry mass by the total cytoplasmic dry mass.

**C. *elegans* brightfield and fluorescence imaging.** For bright field imaging, worms were imaged with an Apo Z 1.5x objective and BF+ light on a Zeiss Axio Zoom V16 dissection scope equipped with an Axiocam 705 mono camera and Zen 3.2 software. Embryos were imaged with bright field illumination on a Leica DM6000B microscope

with a 63x.1.30 HC PL Apo glycerol objective and an Andor iXon Life camera using Visiview (Visitron) software.

To confirm tetraploidy of the 1-4-7 strain, 1% polylysine solution was spotted onto microscope slides and dried. 4 μL M9 buffer were added and 20 adult worms were picked into the drop under a dissection scope. Buffer was carefully removed with a strip of Kimwipes and worms stuck to the polylysine surface. 4 μL of 95% ethanol were added and desiccated. 4 μL 95% Ethanol drops were added again and desiccated and this cycle was repeated four times. 10 μL of Vectashield with 1 μg/mL Hoechst-33342 was pipetted onto the desiccated worms and they were allowed to rehydrate. A 22 mm coverslip was gently placed onto the drop and nail polish was used to seal the coverslip. Fluorescence images of oocyte chromosomes in utero were taken in the DAPI channel on a DeltaVision RT imaging system (Applied Precision, LLC; IX70 Olympus) equipped with a charge-coupled device camera (CoolSNAP HQ; Roper Scientific) in 33 ×0.7 μm z-sections using an Olympus 63×1.34 NA UplanSApo objective. Image stacks were deconvolved using Softworx (Applied Precision, LLC) and maximum-intensity projected for oocyte chromosome planes with FIJI[94]. For counting chromosomes, individual z-planes were assessed in cases of overlapping chromosomes in the maximum-intensity projections.

**FRET imaging.** To measure RanGTP levels in reconstituted nuclei, a YRC probe[53] (final concentration 2 μM) was added to nuclear assembly reactions. Nuclei were sandwiched between coverslips and images were acquired of the CFP and FRET channels. Imaging was performed on a Nikon Widefield Ti2 fluorescence microscope equipped with an sCMOS camera (PCO. edge) and a 40x water immersion objective lens (CFI Apo Lambda S 40XC WI). The system was controlled using the Nikon Elements AR software. Samples were illuminated using LEDs (Lumencor, SpectraX). A 200 ms exposure time was used for acquiring the FRET and CFP images. Image analysis was performed using FIJI[94]. Briefly the FRET and CFP images were divided to obtain an intensity ratio image containing the $I_{FRET}/I_{CFP}$ values. To quantify intensity profiles, line scans (line width: 30 pixels) were performed over a 30 μm length along the major axis of nuclei.

**Nuclear assembly in *Xenopus* egg extracts.** Metaphase-arrested egg extract was prepared from laid *X. laevis* eggs as previously described[95,96]. Briefly, *X. laevis* frogs were primed with 100 U of pregnant mare serum gonadotrophin (PMSG) 3–7 days before the experiment and were boosted with 1000 U human chorionic gonadotrophin (HCG) to induce egg laying. Eggs arrested in the metaphase stage of meiosis II were collected, dejellied using L-Cysteine and fractionated via centrifugation. The cytoplasmic layer was then isolated and supplemented with Cytochalasin and Complete EDTA-free protease inhibitor. Nuclei were assembled in extracts similar to previously published protocols[35]. Extracts were cycled to interphase using $CaCl_2$ (final concentration: 0.6 mM). Cycloheximide (final concentration: 0.1 mg/mL) and energy mix were added to arrest extracts in interphase and support nuclear assembly, respectively. To visualize DNA, membranes and assay nuclear import Hoechst-33342 (final concentration: 0.05 mg/mL), DiI (final concentration: 0.1 mg/mL) and NLS-GFP (final concentration: 0.2 mg/mL) were added. 100 μL extract reactions were prepared for each experimental test. Finally, demembranated sperm nuclei (final concentration: 1000/μL) were added to each tube and incubated in a water bath maintained between 16 and 20 °C. Extracts were mixed every 15 min with the help of a precut pipette tip.

High-speed (HS) extract was prepared as described previously[68]. Metaphase-arrested *Xenopus* egg extracts were first prepared using buffers without EGTA/EDTA. Additionally, cycloheximide (final concentration: 0.1 mg/mL) was added to the extract before the first crushing spin (19,000 × g for 20 min at 4 °C) used to obtain crude extract. Calcium released from the egg lysis step will drive the extract

into interphase. To separate the cytoplasm from the membranes, crude extract was centrifuged at 260,000 × g for 90 min at 4 °C. The cytosolic layer was carefully collected and transferred to centrifuge tubes for a second spin. The second spin was performed at 260,000 × g for 30 min at 4 °C. Aliquots of the final cytosolic layer were then snap frozen in liquid nitrogen and stored at −80 °C. The membrane layer obtained after the first ultra-centrifugation step was washed with a high-salt buffer (1× XB buffer with 0.5 M KCl). Next, the salt-washed membranes were centrifuged through a sucrose cushion (500 mM sucrose in 1× XB buffer), aliquoted, snap frozen in liquid nitrogen and stored at −80 °C. For nuclear assembly reactions, HS extract was slowly thawed on ice. To 100 μL of extract, 5 μL of membrane were added along with energy mix. To visualize DNA, Hoechst-33342 was added (final concentration: 0.05 mg/mL) and to assay nuclear import NLS-GFP (final concentration: 0.2 mg/mL) was added. To initiate the nuclear assembly reaction, *X. laevis* sperm nuclei (final concentration: 1000/μL) were added. Extracts were incubated for 60 min with intermediate mixing every 15 min with a precut pipette tip.

To compare the effects of altering the chromatin content on nuclear size, nuclei were assembled from either *X. laevis* or *X. tropicalis* sperm using the same batch of *X. laevis* interphasic extract. *X. tropicalis* sperm nuclei were obtained from adult *X. tropicalis* male frogs as described previously[97,98].

For imaging, nuclei were sandwiched between two coverslips (631-1573P, VWR International) at different time points. Another coverslip was immediately placed on this coverslip to create a squashed egg extract sample, which provided a thin layer that could be easily imaged with our imaging setup.

**Protein concentration measurements.** Bradford reagent (B 6916, Sigma) was used to determine the concentration of proteins in egg extract and other solutions in this study. As per the manufacturer's recommendation a protein standard curve was first measured using Bovine Serum Albumin (P 0384, Sigma) standards. The standards were dissolved in CSF-XB buffer. To measure protein concentration via the Bradford method, extract was first diluted (100 or 200-fold) in CSF-XB buffer. Next, 50 μL of the protein solution were added to an Eppendorf tube along with 1.5 mL Bradford reagent. Samples were incubated at room temperature for 15 min and subsequently transferred to polystyrene cuvettes (Y195.1, Roth) for absorption measurements. Absorption at 595 nm was measured using an Eppendorf BioSpectrometer Basic. Protein concentrations were determined by matching the absorbance values with the concentration information using the standard curve.

**Calculating the number concentration of nuclear proteins.** To calculate the number concentration of osmotically active protein complexes we used the information available regarding the total nuclear dry mass ($M_{tot}$) and nuclear volume ($V_n$) from our ODT measurements. To calculate the mass of the nuclear proteins ($M_n$) we first subtracted the mass of the chromatin ($M_{chr}$, calculated based on the number of nucleotides and nucleosomes) from the total nuclear mass: $M_n = M_{tot} − M_{chr}$. Next, because we know the average size of protein complexes ($m_n$) in the nucleus (131 kDa[49]), and the nuclear volume ($V_n$) we can calculate the protein complex concentration ($n_n$), i.e. density of the "colloidal" proteins using the equation $n_n = M_n/(m_n V_n)$. This gave us the number concentration of 0.51 mM for nuclear proteins.

**Protein purification.** The GST-NLS-GFP plasmid (pMD49) was a gift from Dr. Thomas Quail (EMBL Heidelberg) and the GST-NLS-GFP protein was purified as described previously[41]. The YFP-RBD-CFP (YRC) chimera protein plasmid (pKW966) was a gift from Prof. Karsten Weis (ETH Zürich), the protein was purified as described previously[53]. Recombinant Nucleoplasmin (Npm2) was purified as described[38].

**Nucleoplasmin immunodepletion and add-back.** To investigate whether Nucleoplasmin2 (Npm2) was necessary for reducing mass density, sperm nuclei in buffer (containing energy mix) were supplemented with recombinant Npm2 (final concentration: 9 μM). The sample was incubated between 16 and 20 °C and imaged at 10 and 30 min after the addition of Npm2. To immunodeplete Npm2 from *Xenopus* egg extract purified polyclonal Npm2 antibody (rabbit α-Npm2 IgG[38]) were first cross linked to Protein A beads (Dynabeads™ Protein A, 10001D, ThermoFisher) using Dimethyl pimelidate dihydrochloride (D8388, Sigma). 80 μg of Npm2 antibody were crosslinked to 330 μL of Protein A beads (10001D, ThermoFisher Scientific). Interphase *Xenopus* egg extract containing cycloheximide was prepared as described above and immunodepletions were performed in a cold room. The extract was subjected to two rounds of immunodepletion and protein reduction levels were assayed using Western blots. Along with immunodepleted samples, a control depletion was performed using the same volume of empty Protein A beads. To test if Npm2 was sufficient for reducing mass density, purified Npm2 was added back to immunodepleted extract to physiological concentrations (4.2 μM[77]).

**Ribosome isolation and add back experiments in *Xenopus* extracts.** 1000 μL of *Xenopus laevis* extract were layered on 33 mL of a 30% sucrose solution in buffer A (20 mM HEPES pH 7.5, 100 mM KCl, 10 mM MgCl$_2$, 5 mM DTT (Dithiothreitol)) in a 70 mL polycarbonate centrifuge tube (Beckman Coulter, No. 355622) and centrifuged for 17 h at 71,000 × g and 4 °C in a 45 Ti fixed-angle rotor (Beckman Coulter, No. 339160). Supernatant was aspirated with a vacuum pump and the ribosome containing pellet was washed twice with Tico buffer (20 mM Hepes pH 7.5, 30 mM KAc, 6 mM MgAc$_2$ and 5 mM DTT). The pellet was resuspended in Tico buffer and the concentration of the ribosome solution was determined to be 100 A$_{260}$ units/mL, using a Nanodrop device.

For ribosome add back experiments, ribosomes were concentrated using Amicon® Ultra-0.5 centrifugal filter devices. Concentrated ribosomes were added back to nuclear assembly (final concentration: 2 μM) reactions in high-speed extracts 60 min after reactions were started. Glycogen was directly added to reactions at a final concentration of ~80 mg/mL[57]. Care was taken to ensure that the extract was not diluted more than 10%. Imaging was performed 10 min after adding additional components.

**Protein gels and Western blots.** Protein samples from each condition were supplemented with SDS buffer and boiled at 95 °C for 10 min. Samples were centrifuged at 1600 × g for 5 min. The supernatant was loaded into wells of a protein gel (NP0301BOX, Invitrogen) with SDS running buffer (NP0001, Invitrogen). Protein bands were separated by running the gel and Instant Blue Coomassie stain (ab119211, abcam) was used to visualise bands. Protein bands were transferred to a PVDF membrane (8858, ThermoFisher) using a transfer buffer (NP0006, Invitrogen). Membranes were blocked using a blocking buffer (5% dry milk in TBST) for 60 min at RT with gentle rocking. Next, the blots were incubated with the primary Npm2 antibody (1:10,000 dilution in blocking buffer) overnight at 4 °C with gentle rocking. A monoclonal αTubulin-antibody (1:15,000 dilution in blocking buffer; T9026, Sigma) was used as a loading control. Blots were washed thrice with TBST with 5 min incubations each. Next the secondary antibody (1:5000 in blocking buffer; anti rabbit-HRP for Npm2 and anti mouse-HRP for αTubulin) was added and incubated for 1 h at room temperature with gentle rocking. The blots were finally washed thrice with TBST with 5 min incubations before adding the ECL substrate solution (1705061, Biorad). Chemiluminescence images were acquired on a Chemidoc™ imaging system (Biorad). Band intensities were quantified using FIJI[94]. Total intensities from the monomeric and pentameric bands were added and normalized to the intensity of the loading control bands.

**Biochemical perturbations.** For biochemical perturbations, inhibitors and drugs were added to the interphasic extracts before the addition of sperm nuclei and the initiation of nuclear assembly. Stock solutions were prepared to ensure that addition of inhibitors did not dilute the extract by more than 10% of the original extract volume. To perturb import, a small molecule inhibitor cocktail of Ivermectin (I8898, Sigma; final concentration: 100 μM) and Pitstop-2 (SML1169, Sigma; final concentration: 30 μM) was used. To prevent chromatin decondensation an inhibitor cocktail of the DNA intercalator Actinomycin D (A1410, Sigma; final concentration: 10 μg/mL), topoisomerase II inhibitor ICRF-193 (I4659, Sigma; final concentration: 150 μM) and the calcium chelator BAPTA (196418, Sigma; final concentration: 5 mM) was supplemented to interphase extracts. To inhibit DNA replication, Aphidicolin (A0781, Sigma; final concentration: 200 μM) was added to the extract before nuclear assembly. To reduce or increase protein concentrations, extracts containing pre-assembled nuclei were either diluted using CSF-XB buffer or protein concentration was increased using a concentrated BSA (A2153, Sigma) solution. Upon addition of the buffer, extract was incubated for 10 min before imaging.

**Osmolality measurements.** A freezing point osmometer (Osmomat 3000 Basic, Gonotec) was used for measuring the osmolality of different extract and protein solutions. 50 μL of sample was used for each assay and at least 2 independent measurements were made for each solution.

## Statistics and reproducibility

In the following section, details about the number of samples, independent experiments, statistical tests and significance values are provided. No statistical test was used to predetermine sample size. No data were excluded from the analyses. All experiments were replicated independently, the number of replicates has been indicated below. Cells and nuclei were imaged randomly. No blinding was carried out as all measurements were quantitative and were not biased by subjective perception. For statistical analysis and plotting, we utilized GraphPad Prism version 9.0 for Mac OS X, GraphPad Software, La Jolla California USA, www.graphpad.com. When necessary, graph visuals such as line thickness, fonts, and colors were optimized using Adobe Illustrator.

**Figure 1** (**b**) n = 15 cells (75 ROIs per species, 5 ROIs per cell). (**c**) n = 15 cells (75 ROIs per species, 5 ROIs per cell). Mann Whitney test was used to test for statistical significance. p value *C. reinhardtii* = 0.0012, p value *S. pombe* < 0.0001, p value *S. cerevisiae* = 0.001, p value *C. elegans* < 0.0001, p value *D. melanogaster* < 0.0001, p value *D. rerio* = 0.01, p value *X. laevis* = 0.02, p value *M. musculus* = 0.009, p value *H. sapiens* = 0.02. In addition to the significance, we also indicate the effect size by calculating Cohen's *d*. An effect size between 0.20–0.50 was considered small, while effect sizes between 0.51–0.80 were considered medium and d > 0.81 was considered large. (**e**) n = 30 nuclei for 0 min, n = 160 nuclei for 5 min, n = 170 nuclei for 15 min, n = 173 nuclei for 30 min and n = 175 nuclei for 60 min from 7 independent experiments. (**f**) n = 18 nuclei from 2 independent experiments. AU = arbitrary units. (**g**) n = 30 nuclei for 0 min, n = 160 nuclei for 5 min, n = 170 nuclei for 15 min, n = 173 nuclei for 30 min and n = 175 nuclei for 60 min from 7 independent experiments.

**Figure 2** (**c**) n = 30 nuclei at 0 min, n = 27 nuclei at 10 min and n = 23 nuclei at 30 min from 3 independent experiments. Mann Whitney test was used to test for statistical significance. p value 0 & 10 min < 0.0001, p value 10 & 30 min < 0.0001. In addition to the significance, we also indicate the effect size by calculating Cohen's *d*. An effect size between 0.20–0.50 was considered small, while effect sizes between 0.51 – 0.80 were considered medium and d > 0.81 was considered large. (**f**) n = 22 nuclei for Ctrl, n = 19 nuclei for Δ and n = 18 nuclei for Δ + Npm2 from 3 independent experiments. Mann Whitney test was used to test for statistical significance. p value Ctrl & Δ = 0.03, p value Δ & Δ + Npm2 = 0.0009 and p value Ctrl & Δ + Npm2 = 0.13.

**Figure 3** (**b**) n = 30 nuclei from 3 independent experiments. Mann Whitney test was used to test for statistical significance. p value +Import & -Import = 0.0001. In addition to the significance, we also indicate the effect size by calculating Cohen's *d*. An effect size between 0.20–0.50 was considered small, while effect sizes between 0.51–0.80 were considered medium and d > 0.81 was considered large. (**c**) n = 30 nuclei for 0 min, n = 30 nuclei for 5 min, n = 30 nuclei for 15 min, n = 30 nuclei for 30 min and n = 25 nuclei for 60 min from 3 independent experiments. Mann Whitney test was used to test for statistical significance. p value +Import & −Import < 0.0001. AU = arbitrary units. (**d**) n = 30 nuclei for 0 min, n = 30 nuclei for 5 min, n = 30 nuclei for 15 min, n = 30 nuclei for 30 min and n = 25 nuclei for 60 min from 3 independent experiments. Mann Whitney test was used to test for statistical significance. p value +Import & −Import < 0.0001. (**e**) n = 30 nuclei for 0 min, n = 30 nuclei for 5 min, n = 30 nuclei for 15 min, n = 30 nuclei for 30 min and n = 25 nuclei for 60 min from 3 independent experiments. Mann Whitney test was used to test for statistical significance. p value +Import & −Import < 0.0001.

**Figure 4** (**c**) The plots in this graph are based on the data that has been reported in Fig. 1e. The predicted volumes from the Theory have been included in the Source Data file. (**d**) n = 29 nuclei for 15 min, n = 29 nuclei for 30 min and n = 30 nuclei for 60 min from 3 independent experiments. (**e**) n = 15 nuclei for 0 min, n = 25 nuclei for 5 min, n = 29 nuclei for 15 min, n = 29 nuclei for 30 min and n = 30 nuclei for 60 min from 3 independent experiments. Mann Whitney test was used to test for statistical significance. p value Laevis & Trops = 0.2750. In addition to the significance, we also indicate the effect size by calculating Cohen's *d*. An effect size between 0.20–0.50 was considered small, while effect sizes between 0.51–0.80 were considered medium and d > 0.81 was considered large. (**f**) n = 39 nuclei from 2 independent experiments. (**g**) n = 15 nuclei for 0 min, n = 25 nuclei for 5 min, n = 29 nuclei for 15 min, n = 29 nuclei for 30 min and n = 30 nuclei for 60 min from 3 independent experiments. Mann Whitney test was used to test for statistical significance. p value Laevis & Trops = 0.0025.

**Figure 5** (**b**) n = 60 cells from 3 independent experiments (300 ROIs, 5 ROIs per cell). Mann Whitney test was used to test for statistical significance. p value Con & Pal < 0.0001, p value Con & Rec = 0.5182. In addition to the significance, we also indicate the effect size by calculating Cohen's *d*. An effect size between 0.20–0.50 was considered small, while effect sizes between 0.51–0.80 were considered medium and d > 0.81 was considered large. (**d**) n = 56 nucleoli for Con and n = 59 nucleoli for Pal from 3 independent experiments. Mann Whitney test was used to test for statistical significance. p value Con & Pal < 0.0001. (**e**) n = 56 nucleoli for Con and n = 59 nucleoli for Pal from 3 independent experiments. Mann Whitney test was used to test for statistical significance. p value Con & Pal < 0.0001. (**f**) For each condition 2 profiles were analysed and each condition was repeated twice. $6 \times 10^6$ cells were used to generate each profile. (**i**) n = 36 nucleoli for both conditions from 3 independent experiments. Mann Whitney test was used to test for statistical significance. p value Con & Pal < 0.0001. AU = arbitrary units. (**j**) n = 36 nucleoli for both conditions from 3 independent experiments. Mann Whitney test was used to test for statistical significance. p value Con & Pal < 0.0001. AU = arbitrary units. (**l**) n = 30 nuclei for 5 min, n = 29 nuclei for 15 min, n = 29 nuclei for 30 min and n = 30 nuclei for 60 min from 3 independent experiments. Mann Whitney test was used to test for statistical significance. p value LS & HS = 0.8697. (**m**) n = 30 nuclei for 5 min, n = 29 nuclei for 15 min, n = 29 nuclei for 30 min and n = 30 nuclei for 60 min from 3 independent experiments. Mann Whitney test was used to test for statistical significance. p value LS & HS = 0.0019. (**o**) n = 175 nuclei for LS from 7 independent experiments, n = 30 nuclei for HS from 3 independent experiments, n = 10 nuclei for H + R from 2 independent experiments and n = 30 nuclei for HS + R + G from 3 independent experiments. Mann Whitney test was used to test for statistical significance. p value LS & HS < 0.0001 and p value LS & HS + R + G = 0.1929.

**Figure 6** (**b**) n = 24 nuclei and cells for 2 cell stage, n = 48 nuclei and cells for 4 cell stage, n = 79 nuclei and cells for 8 cell stage and n = 85 nuclei and cells for 16 cell stage for wildtype (2n) embryos. n numbers are the same for panels d, f, g, j, l. (**c**) n = 20 nuclei and cells for 2 cell stage, n = 44 nuclei and cells for 4 cell stage, n = 80 nuclei and cells for 8 cell stage and n = 58 nuclei and cells for 16 cell stage for tet: 1-4-7 (4n) embryos. n numbers are the same for panels e, h, i, k, l. (**f**) Mann Whitney test was used to test for statistical significance. p value 2 cell & 4 cell = 0.9834, p value 4 cell & 8 cell =0.7824, p value 8 cell & 16 cell = 0.9503. (**g**) Mann Whitney test was used to test for statistical significance. p value 2 cell & 4 cell = 0.2819, p value 4 cell & 8 cell = 0.0771, p value 8 cell & 16 cell = 0.1000. (**h**) Mann Whitney test was used to test for statistical significance. p value 2 cell & 4 cell = 0.9688, p value 4 cell & 8 cell = 0.7481, p value 8 cell & 16 cell = 0.0675. (**i**) Mann Whitney test was used to test for statistical significance. p value 2 cell & 4 cell = 0.9256, p value 4 cell & 8 cell = 0.9772, p value 8 cell & 16 cell = 0.0751. (d, e, j-l) $R^2$ values indicated for each fit.

### Reporting summary

Further information on research design is available in the Nature Portfolio Reporting Summary linked to this article.

### Data availability

All data needed to evaluate the conclusions are present in the paper or the supplementary information and are available from the corresponding author upon request. Source data are provided with this paper.

### Code availability

The code utilized in this study to reconstruct RI tomograms is publicly available online at https://github.com/OpticalDiffractionTomography/ODT_FieldTomogramGUI.

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

## Acknowledgements

The authors acknowledge funding from the DFG (RE 3925/1-1 and 528483508 – FIP 12 to S.R., SPP2191 KR 5755/1-1 to M.L.K. and NI 1151/2-1 to R.N.), the National Institutes of Health (R01GM135614 to D.S.), the Joachim-Herz-Stiftung (to A.B.) and the Max Planck Society (to J.G., V.Z., S.R.). The authors would like to thank all current and former members of the Guck, Reber, and Zaburdaev labs. We thank Dr. Roland Knorr (HU Berlin) and Dr. Alexander May (Tokyo Institute of Technology) for the S. cerevisiae cells, Auguste Palm (FU Berlin) and Prof. Dr. Ann Ehrenhofer Murray (HU Berlin) for the S. pombe cells, Dr. Olga Baidukova and Prof. Dr. Peter Hegemann (HU Berlin) for the C. reinhardtii cells, Prof. Dr. Wolfram Antonin (RWTH Aachen) for the XL-177 cells, Dr. Alf Herzig (MPIIB-Berlin) for the D. melanogaster cells, Anika Neuschulz and Dr. Jan Philipp Junker (MDC Berlin) for the D. rerio cells. We would like to thank Dr. Thomas Quail (EMBL Heidelberg) and Prof. Dr. Karsten Weis (ETH Zurich) for the GST-NLS-GFP and YRC plasmids, respectively. We also thank Tobias Kletter (MPI-IB) for help with figure design, Prof. Dr. Simon Alberti and Dr. Anna Taubenberger (TU-Dresden) for permission to use their correlative fluorescence-ODT setup, Prof. Dr. Frank Buccholz and Martina Augsburg for sharing the RPE1 cells, and Dr. Jan Schromanzer and his team at the Advanced Medical BIOimaging Core Facility, Charité Berlin. We thank Marcus Taylor (MPI-IB), Dirk Görlich (MPI-Nat), Mathieu Piel and Romain Rollin (Institut Curie) for helpful comments on the manuscript.

## Author contributions

This work represents a truly collaborative effort. Each author has contributed significantly to the findings and regular group discussions guided the development of the ideas presented here. In more detail, A.B. performed all experiments and analyzed the data. O.M. and V.Z. conceptualized and wrote the theory part. K.K. designed the ODT setup and wrote the script for controlling the setup. A.B. and K.K. built the correlative fluorescence-ODT setup. K.K. developed the script for field retrieval, tomogram reconstruction and axial drift correction. A.B., K.K. and C.H. performed the *C. elegans* experiments. B.L. and D.S. purified the recombinant Npm2 and Npm2 antibodies. M.K. and R.N. provided the antibodies for the immunofluorescence and performed the ribosome profile analysis. R.N. purified the ribosomes used for addback experiments. O.M., K.K., V.Z., D.S. and J.G. contributed to experimental design, data analysis, and interpretation. A.B., J.G., V.Z,. and S.R. conceived the project. S.R. wrote the manuscript with input from all authors.

## Funding

## Competing interests

The authors declare no competing interests.
