## [Transparent Peer Review file · Nature Communications]

Conserved nucleocytoplasmic density homeostasis drives cellular organization across eukaryotes

Corresponding Author: Professor Simone Reber

Version 0:

Reviewer comments:

Reviewer #1

(Remarks to the Author)

In this extensively revised and expanded manuscript, the authors provide additional evidence for NC density homeostasis and revealed further underlying principles. I previously reviewed this manuscript for Nature Cell Biology. Overall, the study provides perhaps the most comprehensive characterization to date of mass density homeostasis between the nucleus and cytosol. In response to my concern about the extent to which they experimentally perturb their systems, they now include substantive new data on the following: (1) further modulation of cytoplasmic density in model for senescence (new Fig 5), (2) osmotic stress treatment (Extended Data Fig 5), (3) effect of DNA replication inhibition on density ratios (Extended Data Fig 4H). They specifically address my comment about functional significance by perturbing cytoplasmic and nuclear density via osmotic shock and in conditions that mimic senescence. They add cell line data for *Xenopus*, and show variation in NC density ratio for cancer cell lines with larger sizes, which in turn appears similar to senescent-like cell in which NC density ratio homeostasis is lost (shifts from ~ 0.8 to > 1) (new Fig 5b). They tie this senescence phenotype to reduced ribosome production and enrichment of pre-ribosomal subunits in nucleoli. Further I feel they now better explain in text and schematize in figures the sources of osmotic pressure that lead to lower nucleoplasm density compared to the cytosol, include a table of contribution of DNA to osmotic pressure (Extended Data Table 1) and provide experimental evidence that dense cytosolic components (glycogen and ribosome) keep NC density ratio < 1 (Fig 6n,o). As mentioned in my first review, the use of optical diffraction tomography (ODT) is a clever method to determine refractive index which is proportional to dry mass density and this methodology can be broadly useful in many other contexts.

I have one remaining comment. The authors claim the nucleus behaves like a “near-ideal osmometer” based in part on the new osmotic shock experiments. However, the data in Extended Data Fig 5 indicates that at higher cytosolic densities that NC ratio density < 1 is lost; and their experiments diverge from the theory. I may have missed it but I believe it important for the authors to explain this disparity. Note, I also did not see the mouse cell line osmotic shock data shown in the rebuttal. This data suggested under hyperosmotic conditions that density of cytosol and nucleus increased but overall NC density ratio was largely unchanged and well below 1 (again different from what was shown in Extended Data Fig 5 in vitro).

(Remarks on code availability)

Reviewer #4

(Remarks to the Author)

The main observation by Biswas et al., that the nucleo-cytoplasmic (NC) density ratio is below one (~ 0.8) and remains constant across evolution (in nine different organisms) and during development (in *C. elegans* embryos), is both convincing and intriguing. The experimental work is carefully executed and the proposed model, along with the hypothesis of a universal regulation of the NC density ratio, is likely to stimulate significant discussion in the field.

Apart from quantitative discrepancies probably due to the different systems used for the estimates (*Xenopus* egg vs human

cells), the main point of the model, that chromatin entropy gives a small contribution to the overall osmotic pressure of the nucleus, actually qualitatively agrees with Deviri and Safran findings (this is evident from the answer to reviewer #2). The lengthy discussion on quantitative changes on models that rely on arbitrary simplifications and assumptions is probably mainly a distraction of the readers and should be drastically simplified.

The model is interesting, but complex, and contains many ingredients and parameters, as well as not tested (or testable) assumptions. In the end the agreement based on three adjustable parameters is not impressive. Also, it is hard to tell apart statements that are strictly backed up by data from educated opinions.

Please find below some suggestions that may help the authors to improve the manuscript.

Major:

1) Refuting simplified parameter-poor models and reconstructing their narrative based on this broken-down analyses. These can be stripped-down versions of their full model. At least these analyses can rationalize why they think all the model ingredients are essential. For example (i) Can a model without the contribution of chromatin entropy be refuted (probably not)? (ii) can a model without the direct AND indirect contribution of chromatin entropy be refuted (probably yes and this is the central point of the paper)? (iii) can a model without protein complexes be refuted (probably yes and this is a minor point of the paper, and probably a chromatin-less model is sufficient to support this point) ? etc...

2) Performing comparisons at everything else equal. For example they compare their quantitative estimate of entropic chromatin pressure with that of Deviri and Safran as frog vs human, but we are more interested in learning how the models compare *on the same system*.

3) Another question is whether the indirect contribution of chromatin through nuclear import-export bias (as a regulator of nuclear import via the RanGTP gradient) can be actually described mechanistically as something else (NE tension-biased transport ?). It is a bit disturbing to consider nuclear import solely controlled by chromatin. Experimentally the authors didn't rule out that the nuclear-cytoplasmic transport bias comes from other biological processes than chromatin, so maybe cautionary sentences could be of use as well as mention alternatives. This would not impact the main conclusions of the work.

Minor:

1) Although the NC density ratio appears to be constant across the nine different organisms, the absolute difference in density between the nucleus and cytoplasm seems to decrease in mammals (mouse and human). Is this observation correct? It would be helpful if the authors could comment on this apparent trend.

2) We are not sure about it, but, in terms of phylogeny, shouldn't fish precede amphibians (frog)? Could be this a more correct order: *C. reinhardtii*, *S. pombe*, *S. cerevisiae*, *C. elegans*, *D. melanogaster*, *D. rerio*, *X. laevis*, *M. musculus*, *H. sapiens* ?

3) The model is mainly designed thinking to fertilized egg. In this scenario the cytoplasm is approximated to an uniform "soup". However, as well for the nucleus, it should be considered that also in the cytoplasm not all the molecules counted for density are equally participating to the generation of colloid osmotic pressure (as many of them are indeed assembled in macro-structures as micro-tubules or actin fibers or similar). Could the authors comment on this point?

(Remarks on code availability)

Reviewer #5

(Remarks to the Author)

The main observation by Biswas et al., that the nucleo-cytoplasmic (NC) density ratio is below one (~0.8) and remains constant across evolution (in nine different organisms) and during development (in *C. elegans* embryos), is both convincing and intriguing. The experimental work is carefully executed and the proposed model, along with the hypothesis of a universal regulation of the NC density ratio, is likely to stimulate significant discussion in the field.

Apart from quantitative discrepancies probably due to the different systems used for the estimates (*Xenopus* egg vs human cells), the main point of the model, that chromatin entropy gives a small contribution to the overall osmotic pressure of the nucleus, actually qualitatively agrees with Deviri and Safran findings (this is evident from the answer to reviewer #2). The lengthy discussion on quantitative changes on models that rely on arbitrary simplifications and assumptions is probably mainly a distraction of the readers and should be drastically simplified.

The model is interesting, but complex, and contains many ingredients and parameters, as well as not tested (or testable) assumptions. In the end the agreement based on three adjustable parameters is not impressive. Also, it is hard to tell apart statements that are strictly backed up by data from educated opinions.

Please find below some suggestions that may help the authors to improve the manuscript.

Major:

1) Refuting simplified parameter-poor models and reconstructing their narrative based on this broken-down analyses. These can be stripped-down versions of their full model. At least these analyses can rationalize why they think all the model ingredients are essential. For example (i) Can a model without the contribution of chromatin entropy be refuted (probably not)? (ii) can a model without the direct AND indirect contribution of chromatin entropy be refuted (probably yes and this is the central point of the paper)? (iii) can a model without protein complexes be refuted (probably yes and this is a minor point of the paper, and probably a chromatin-less model is sufficient to support this point) ? etc...

2) Performing comparisons at everything else equal. For example they compare their quantitative estimate of entropic chromatin pressure with that of Deviri and Safran as frog vs human, but we are more interested in learning how the models compare *on the same system*.

3) Another question is whether the indirect contribution of chromatin through nuclear import-export bias (as a regulator of nuclear import via the RanGTP gradient) can be actually described mechanistically as something else (NE tension-biased transport ?). It is a bit disturbing to consider nuclear import solely controlled by chromatin. Experimentally the authors didn't rule out that the nuclear-cytoplasmic transport bias comes from other biological processes than chromatin, so maybe cautionary sentences could be of use as well as mention alternatives. This would not impact the main conclusions of the work.

Minor:

1) Although the NC density ratio appears to be constant across the nine different organisms, the absolute difference in density between the nucleus and cytoplasm seems to decrease in mammals (mouse and human). Is this observation correct? It would be helpful if the authors could comment on this apparent trend.

2) We are not sure about it, but, in terms of phylogeny, shouldn't fish precede amphibians (frog)? Could be this a more correct order: *C. reinhardtii*, *S. pombe*, *S. cerevisiae*, *C. elegans*, *D. melanogaster*, *D. rerio*, *X. laevis*, *M. musculus*, *H. sapiens* ?

3) The model is mainly designed thinking to fertilized egg. In this scenario the cytoplasm is approximated to an uniform "soup". However, as well for the nucleus, it should be considered that also in the cytoplasm not all the molecules counted for density are equally participating to the generation of colloid osmotic pressure (as many of them are indeed assembled in macro-structures as micro-tubules or actin fibers or similar). Could the authors comment on this point?

(Remarks on code availability)

Version 1:

Reviewer comments:

Reviewer #1

(Remarks to the Author)

This further revised manuscript satisfies my few remaining minor concerns. In my opinion it is ready for publication

(Remarks on code availability)

Reviewer #4

(Remarks to the Author)

I am happy with the revisions, and I think the manuscript should be accepted. I would only encourage the authors to use a bit more cautionary sentences in main text and SI concerning the model conclusions. They addressed all the points raised, but parameters remain an issue and many of the falsifications arguments they produced rely on quantitative discrepancies that are not very large, and could be parameter-dependent. The debate on such models remains open and while the authors' contribution is very valuable, it is likely new data will help us draw a clearer picture in the coming years: in this perspective it is in the interest of everyone to formulate robust statements about current results.

(Remarks on code availability)

Reviewer #5

(Remarks to the Author)

The authors made a significant effort to address my concerns (as well as those of the other reviewers), which is greatly appreciated.

They did an excellent job in thoroughly answering and discussing the raised questions.

I therefore fully endorse the manuscript for publication.

(Remarks on code availability)

Reviewer #1 (Remarks to the Author)

In this extensively revised and expanded manuscript, the authors provide additional evidence for NC density homeostasis and revealed further underlying principles. I previously reviewed this manuscript for Nature Cell Biology. Overall, the study provides perhaps the most comprehensive characterization to date of mass density homeostasis between the nucleus and cytosol. In response to my concern about the extent to which they experimentally perturb their systems, they now include substantive new data on the following: (1) further modulation of cytoplasmic density in model for senescence (new Fig 5), (2) osmotic stress treatment (Extended Data Fig 5), (3) effect of DNA replication inhibition on density ratios (Extended Data Fig 4H). They specifically address my comment about functional significance by perturbing cytoplasmic and nuclear density via osmotic shock and in conditions that mimic senescence. They add cell line data for *Xenopus*, and show variation in NC density ratio for cancer cell lines with larger sizes, which in turn appears similar to senescent-like cell in which NC density ratio homeostasis is lost (shifts from ~ 0.8 to > 1) (new Fig 5b). They tie this senescence phenotype to reduced ribosome production and enrichment of pre-ribosomal subunits in nucleoli. Further I feel they now better explain in text and schematize in figures the sources of osmotic pressure that lead to lower nucleoplasm density compared to the cytosol, include a table of contribution of DNA to osmotic pressure (Extended Data Table 1) and provide experimental evidence that dense cytosolic components (glycogen and ribosome) keep NC density ratio < 1 (Fig 6n,o). As mentioned in my first review, the use of optical diffraction tomography (ODT) is a clever method to determine refractive index which is proportional to dry mass density and this methodology can be broadly useful in many other contexts.

We thank reviewer #1 for taking the time to re-review the manuscript in detail. We are delighted that they think it provides the most comprehensive characterization of mass density homeostasis.

(1) I have one remaining comment. The authors claim the nucleus behaves like a "near-ideal osmometer" based in part on the new osmotic shock experiments. However, the data in Extended Data Fig 5 indicates that at higher cytosolic densities that NC ratio density < 1 is lost; and their experiments diverge from the theory. I may have missed it but I believe it important for the authors to explain this disparity.

We agree with Reviewer #1 that at very high cytoplasmic densities ($\rho > 200$ mg/mL) nuclear size deviates from a linear response to cytoplasmic density and thus the near-ideal osmometer idea. One possible explanation for this is a lower limit to nuclear size at which non-ideal interactions can no longer be neglected. In such a scenario our

model, which considers macromolecules as an ideal gas, will naturally deviate from an ideal osmometer description. We had previously addressed this discrepancy in Supplementary Notes lines 964 - 968 as follows:

“The reason for the larger discrepancy in the hyper-osmotic shock data is most likely a consequence of the smaller size and much higher density of nuclei in this case, which makes it a highly non-ideal system. This makes our prediction, which assumes an ideal form for the colloid osmotic pressure, less accurate.”

We now mention this discrepancy earlier in the Supplementary Notes, at the point where we introduce the idea of an ideal osmometer, see lines 295 - 299 as follows:

“We can also neglect nuclear envelope tension as supported by Supplementary Notes Fig. 8 where we show that the *X. laevis* nucleus behaves like a near-ideal osmometer under hypo-osmotic and control conditions (section 1.4.5). Under strongly hyper-osmotic conditions, the high density of the nucleus leads to deviations from ideal gas law.”

Taken together, we no longer mention the near-ideal osmometer idea in the main text. It is now only discussed for the interested readers in the Supplementary Notes.

(2) Note, I also did not see the mouse cell line osmotic shock data shown in the rebuttal. This data suggested under hyperosmotic conditions that density of cytosol and nucleus increased but overall NC density ratio was largely unchanged and well below 1 (again different from what was shown in Extended Data Fig 5 *in vitro*).

Reviewer #1 is correct in that we have not included the mouse cell line osmotic shock data. This is because very similar experiments have been published (Finan et al. 2009, Kim et al. 2016, Lemière et al. 2022) and the scaling of nuclear size with increasing cell volumes is a well understood phenomenon (Webster et al. 2009, Rollin et al. 2023). We are happy to include the density data at the editor’s discretion.

In the *in vitro Xenopus* experiment with 100 mg of additional protein added directly to the cytoplasm we induce a much stronger osmotic challenge as compared to the addition of 50 mM Sorbitol to the medium of mouse cells as indicated by the overall nuclear volume changes (Figure on page 10 of previous rebuttal letter and Supplementary Fig. 5). Moreover, the behavior of mouse nuclei in a cellular context is more complex and is best described by a “nested Pump-Leak model” (Rollin et al. 2023). Briefly, the mouse cells are a system of two embedded compartments (nucleus in the cell), whereas the *X. laevis* nucleus is a single compartment within an effectively infinite pool of cytoplasm (extract).

Reviewer #4 and #5 (Remarks to the Author):

The main observation by Biswas et al., that the nucleo-cytoplasmic (NC) density ratio is below one (~ 0.8) and remains constant across evolution (in nine different organisms) and during development (in *C. elegans* embryos), is both convincing and intriguing. The experimental work is carefully executed and the proposed model, along with the hypothesis of a universal regulation of the NC density ratio, is likely to stimulate significant discussion in the field.

We thank reviewer #4 and #5 for their positive assessment of our work.

Apart from quantitative discrepancies probably due to the different systems used for the estimates (*Xenopus* egg vs human cells), the main point of the model, that chromatin entropy gives a small contribution to the overall osmotic pressure of the nucleus, actually qualitatively agrees with Deviri and Safran findings (this is evident from the answer to reviewer #2).

We fully agree that our model qualitatively agrees with the work by Deviri and Safran in the description of the pressure balance. Where our model goes beyond previous models, however, is by dynamically coupling pressure balance, nuclear import, and nuclear growth.

The lengthy discussion on quantitative changes on models that rely on arbitrary simplifications and assumptions is probably mainly a distraction of the readers and should be drastically simplified.

We made a significant effort to streamline the model presentation and make it as concise as possible in the main text. We would suggest keeping the detailed description for the interested readers in the Supplemental Notes, in which we address and discuss all the assumptions and simplifications.

The model is interesting, but complex, and contains many ingredients and parameters, as well as not tested (or testable) assumptions. In the end the agreement based on three adjustable parameters is not impressive. Also, it is hard to tell apart statements that are strictly backed up by data from educated opinions.

In the Supplemental Notes, we now better highlight the parameters that have been directly measured, derived from measured quantities, and the ones that have been taken from literature (see Supplementary Notes Table 2). For the reviewer, we include a table below that shows all the parameters needed to fully describe our model equations

and their values (correspond to the rows in bold in Supplementary Notes Table 2 and colored columns in Supplementary Notes Table 3).

Quantity	Value	Unit	Note
Inward protein transport rate k_{in}	1.43 ± 0.07	$\frac{10^{-3}}{s}$	Free Parameter
Chromatin compaction parameter κ	5.1 ± 0.6	$10^{-4} \mu m$	Free Parameter
Transport enhancement parameter v_f	1.15	-	Free Parameter
Cytoplasmic density ρ_c	100 ± 2	mg/mL	Our measurements
Total nuclear dry mass (60 min) M_{tot}	51 ± 2	pg	Our measurements
Outward protein transport rate k'_{out}	3.18 ± 0.19	$\frac{10^{-6}}{s \mu m^3}$	Set by measurements
Excluded volume of chromatin V_{chr}	48 ± 4	μm^3	Set by measurements
Number avg. native weight of nuclear proteins m_n	131	kDa	Frey, et al. 2018
Number avg. native weight of cytoplasmic proteins m_c	155	kDa	Frey, et al. 2018
Genome length (Xl) N_{bp}	3.1	Gbp	Hellsten, et al. 2010
Genome length (Xt) N_{bp}	1.7	Gbp	Hellsten, et al. 2010
Nucleosome repeat length l_{repeat}	200	bp	Halverson, et al. 2014
Avg. mass of basepair m_{bp}	0.65	kDa	Bionumbers, BNID 110968
Mass of H2.A (Xl) m_{h2a}	14	kDa	UniProt

Mass of H2.B (XI) m_{h2b}	14	kDa	UniProt
Mass of H3 (XI) m_{h3}	15	kDa	UniProt
Mass of H4 (XI) m_{h4}	11	kDa	UniProt
Mass of B4 linker histone (XI) m_{h1m}	29	kDa	UniProt

As can be seen, only 3 out of the 18 model parameters are adjustable and the rest is either determined by our own measurements or reliable values taken from literature. Moreover, the fitted values of two of our adjustable parameters are remarkably consistent with values found in literature: the fitted value for the inward protein transport rate sets a time scale in the order of 10^3 seconds for protein transport to reach steady state, which is consistent with (Kopito and Elbaum, 2007). Further, the fitted value for the chromatin compaction parameter corresponds to a range of values for the chromatin persistence length and linear density that is consistent with the literature (Dekker and Steensel 2013, Halverson et al. 2014, Sugawara and Kimura 2017), see Supplementary Notes section 1.5.3 and Table 2. Additionally, we directly measured the cytoplasmic density and nuclear dry mass, which further refined the accuracy of our model. Taken together, most of the parameter values of our model are backed up by reliable data.

Please find below some suggestions that may help the authors to improve the manuscript.

Major:

1) Refuting simplified parameter-poor models and reconstructing their narrative based on this broken-down analyses. These can be stripped-down versions of their full model. At least these analyses can rationalize why they think all the model ingredients are essential. For example (i) Can a model without the contribution of chromatin entropy be refuted (probably not)? (ii) can a model without the direct AND indirect contribution of chromatin entropy be refuted (probably yes and this is the central point of the paper)? (iii) can a model without protein complexes be refuted (probably yes and this is a minor point of the paper, and probably a chromatin-less model is sufficient to support this point) ? etc...

We thank the reviewer for this important comment. In fact, this is exactly how we proceeded with the development of our model. Importantly, most of these points were addressed in the original submission. Now, we provide a detailed explanation with

additional experimental data below:

(i) Can a model without the contribution of chromatin entropy be refuted?

We directly show the importance of the entropic chromatin pressure by comparing colloid osmotic pressure only (see Figure below, $\Delta P_{\text{proteins}}$, squares) and the combination of colloid osmotic pressure and entropic chromatin pressure ($\Delta P_{\text{proteins}} + P_{\text{chromatin}}$, diamonds). Especially at 60 minutes, the predictions without entropic chromatin pressure significantly underestimate nuclear volumes in both *X. laevis* and *X. tropicalis*. Thus, predictions that do consider entropic chromatin pressure more accurately reproduce experimental data. These data (as shown below) are provided as Figure 4c & 4d and described in the main text, lines 201 - 203, 219 - 221.

Chromatin pressure improves the accuracy of volume predictions

(a) Volume of *X. laevis* nuclei increases over time. Experimental data points (blue solid circles and bars show mean \pm SEM) and predicted values from pressure balance. Considering colloid osmotic pressure by proteins (squares) alone results in an underestimation of nuclear volume. Including chromatin pressure (diamonds) matches experimental values. **(b)** Volume of *X. tropicalis* nuclei increases over time. Experimental data points (red solid circles and bars show mean \pm SEM) and predicted values from pressure balance. Again, considering colloid osmotic pressure by proteins (squares) alone results in an underestimation of nuclear volume. Including chromatin pressure (diamonds) matches experimental values.

(ii) Can a model without the direct AND indirect contribution of chromatin entropy be refuted?

To demonstrate that both the direct and indirect contribution of chromatin are required for nuclear growth and final density establishment, we performed nuclear assembly in extracts containing an inhibitor combination of ActD, ICRF-193 and BAPTA. ActD and ICRF-193 inhibit chromatin decondensation and topoisomerase activity (Guy and Taylor 1978, Ishida et al. 1994) thus reducing direct effects of chromatin pressure. BAPTA forms poreless nuclei (without NPCs; Macaulay and Forbes 1996), therefore these nuclei are import-deficient and the indirect effect of chromatin is abrogated. Importantly, abolishing the direct and indirect contribution of chromatin results in the formation of

nuclei that remain small in size, do not accumulate dry mass, and have a high density comparable to that of fully condensed sperm chromatin. This experimental evidence is included as Supplementary Fig. 3 and mentioned in the main text, lines 175 - 178 as follows: “Consistently, when we simultaneously inhibited chromatin decondensation and nuclear pore formation, nuclei retained a density as high as that of condensed sperm chromatin (Supplementary Fig. 3a-d).”

Inhibition of chromatin decondensation and nuclear import leaves nuclei with a density comparable to that of fully condensed sperm chromatin.

(a) Nuclei were assembled in *Xenopus* egg extracts in the presence of ICRF193 (a topoisomerase inhibitor), Actinomycin-D (ActD, a DNA intercalator) and BAPTA (a calcium chelator). This inhibitor cocktail inhibits chromatin decondensation and allows for poreless nuclear envelope closure. Top panel shows representative fluorescence images of Hoechst-33342 stained nuclei (DNA) from different time points. Bottom panel shows the corresponding RI image. While assembling nuclei round up, they have a RI value comparable to that of fully condensed sperm chromatin.

(b) Volume (V_n , grey) of poreless nuclei. Symbols represent the mean values. In b-d, circles and lighter colors show the values for the control nuclei while squares and darker colors show the values for perturbed nuclei. $n = 20$ from 2 independent experiments. At 60 minutes there is a significant difference between the volumes of nuclei. In b-d, dashed lines are used to guide readers along the trend followed by the experimental data.

(c) Dry mass (M_n , yellow) of poreless nuclei. For poreless nuclei, M_n does not change during nuclear assembly.

(d) Density of poreless nuclei (ρ_n , purple). For poreless nuclei, ρ_n does not change during nuclear assembly. Black dashed line: ρ of the cytoplasm.

Black lines and bars in all graphs represent the mean \pm SEM. Mann-Whitney test where **** indicates $p < 0.0001$. Cohen's d indicated in each graph with statistical significance.

Further experimental evidence for the combined direct and indirect effect of chromatin comes from the replication experiments, where a duplication of DNA content not only leads to an increase in nuclear volume but also an increase in dry mass via a higher protein import (see Supplementary Fig. 4 d-h).

(iii) Can a model without protein complexes be refuted?

We thank the reviewer for this comment. One important advance of our work is that we have directly measured the protein content in both the nucleus and the cytoplasm. Together with the data of Wühr et al. 2015, this allows us to calculate the colloid osmotic pressure for protein complexes in the nucleus and the cytoplasm.

Now, we included the same calculations assuming individual proteins with no complex formation. This results in pressures that lead to significantly smaller nuclear sizes than observed in experiments (see figure below). Even adding chromatin pressure does not significantly improve the predictions. Thus, only by considering the colloid pressure of protein complexes consistently describes the nuclear densities and volumes as observed experimentally. We now include these estimates in Supplementary Notes Table 2, Supplementary Notes Fig. 2 as seen below and refer to them in the main text, lines 201 - 203 as follows: "Accounting for protein complexes significantly lowered the number estimate with consequences for colloid osmotic pressures (Supplementary Notes Fig. 2 and Table 2)."

Comparison of nuclear volume predictions.

(a) Volume predictions assuming no protein complex formation in *X. laevis*.

(b) Volume predictions assuming protein complex formation in *X. laevis*.

(c) Volume predictions assuming no protein complex formation in *X. tropicalis*.

(d) Volume predictions assuming protein complex formation in *X. tropicalis*.

In each graph, solid circles show the experimental data, squares show the volume predictions obtained from a pressure balance considering only the colloid osmotic pressure from proteins and diamonds show the predictions obtained from a balance with colloid osmotic pressure and chromatin pressure, as depicted in Supplementary Equation (5).

Pressure at 60 min	X. laevis	X. tropicalis
Cytoplasmic Colloid Osmotic Pressure (Complexes, $m_c = 155$ kDa)	1.61 ± 0.05 kPa	1.61 ± 0.05 kPa
Cytoplasmic Colloid Osmotic Pressure (No Complexes, $m_c = 39$ kDa)	6.4 ± 0.2 kPa	6.4 ± 0.2 kPa

Nuclear Colloid Osmotic Pressure (Complexes, $m_n = 131$ kDa)	1.28 ± 0.04 kPa	1.38 ± 0.07 kPa
Nuclear Colloid Osmotic Pressure (No Complexes, $m_n = 47$ kDa)	3.57 ± 0.10 kPa	3.9 ± 0.2 kPa
Chromatin Pressure	0.54 ± 0.06 kPa	0.28 ± 0.07 kPa

2) Performing comparisons at everything else equal. For example they compare their quantitative estimate of entropic chromatin pressure with that of Deviri and Safran as frog vs human, but we are more interested in learning how the models compare *on the same system*.

As mentioned above, we in fact use the “human” model of Deviri and Safran to describe the steady-state nuclear volume using parameters measured in *Xenopus* extracts. The advance here, however, is that we directly measure colloid osmotic pressure by protein complexes.

To make the model more quantitative for human cells, we would not only require information on protein complex sizes but also on nuclear and cytoplasmic protein concentrations. For the *Xenopus* system, Wühr et al, 2015 provide specific protein concentrations that we use for weighted averages together with our density measurements (Supplementary Notes Table 2). Proteomic studies such as Havugimana et al. 2013, Connelly et al. 2019, and Drew et al. 2021 provide information on protein complexes, however, the distinct size of the complexes in the nucleus and cytoplasm, and protein concentration measurements in each compartment are lacking. To our knowledge, *Xenopus* is the only system for which such quantitative and systematic data is complete and available.

3) Another question is whether the indirect contribution of chromatin through nuclear import-export bias (as a regulator of nuclear import via the RanGTP gradient) can be actually described mechanistically as something else (NE tension-biased transport ?). It is a bit disturbing to consider nuclear import solely controlled by chromatin. Experimentally the authors didn't rule out that the nuclear-cytoplasmic transport bias comes from other biological processes than chromatin, so maybe cautionary sentences could be of use as well as mention alternatives. This would not impact the main conclusions of the work.

We thank the reviewers for bringing up this point. While our experimental data, the measurements of the NLS-GFP accumulation and the RanGTP gradient in *X. laevis* vs. *X. tropicalis* (see Supplementary Fig. 4b and Fig. 4f) points towards a significant effect of chromatin amount on protein import, additional factors such as mechanical forces can affect nuclear import. However, in the *Xenopus* egg extract nuclear envelope tension is unlikely to be such a factor because membrane components are not limiting (Mes-Hartree and Armstrong 1976) and mechanical constraints on the nuclear envelope can be neglected (see response to reviewer #1's comments from the previous rebuttal letter and Supplementary Fig. 5). We now explicitly mention this in the main text (see lines 360 - 366) and include appropriate references in the discussion.

Chromatin amount directly influences nuclear import

(a) Quantification of NLS-GFP intensity at different time points of nuclear assembly ($n = 30$ from 3 independent experiments). *X. laevis* nuclei are more import efficient in comparison to *X. tropicalis* nuclei. **(b)** RanGTP levels in *X. laevis* (top) and *X. tropicalis* (bottom) nuclei using a FRET probe. Color scale: fluorescence intensity range. Line scan quantifies FRET ratio signal ($\text{Intensity}_{\text{FRET}}/\text{Intensity}_{\text{CFP}}$), $n = 39$ from 2 independent experiments. FRET ratio signal decreases within nuclei due to the high concentration of RanGTP in comparison to the cytoplasm. Scale bar: 5 μm .

Mann-Whitney test where *** indicates $p < 0.001$. Cohen's d indicated with statistical significance.

Minor:

1) Although the NC density ratio appears to be constant across the nine different organisms, the absolute difference in density between the nucleus and cytoplasm seems to decrease in mammals (mouse and human). Is this observation correct? It would be helpful if the authors could comment on this apparent trend.

While the data depicted in Figure 1 may imply such a trend, we would refrain from claiming this in a very broad and general way. We would need to more systematically and carefully address this question. It is important to note that the absolute cytoplasmic and nuclear densities can also vary significantly between different cell types in one species (see figure below). However, they all maintain a constant NC density ratio of 0.8 ± 0.01 (Mean \pm SEM).

Subcellular densities vary between different human cell types

(a) Nuclear (Nuc, dark purple circles) and cytoplasmic (Cyto, light purple circles) densities (ρ) vary significantly between different human cell types. (b) NC density ratio (ρ_n / ρ_c) is conserved between different cell types. Dashed line shows the average value from all cells. Each circle represents the average density value from a single cell. $n = 40, 69,$ and 37 cells for hAEC, RPE-1 and neutrophils, respectively. Black lines and bars show the Mean \pm SEM.

2) We are not sure about it, but, in terms of phylogeny, shouldn't fish precede amphibians (frog)? Could be this a more correct order: *C. reinhardtii*, *S. pombe*, *S. cerevisiae*, *C. elegans*, *D. melanogaster*, *D. rerio*, *X. laevis*, *M. musculus*, *H. sapiens* ?

We thank reviewer #4 for this hint. We have changed Figure 1 accordingly.

3) The model is mainly designed thinking to fertilized egg. In this scenario the cytoplasm is approximated to an uniform "soup". However, as well for the nucleus, it should be considered that also in the cytoplasm not all the molecules counted for density are equally participating to the generation of colloid osmotic pressure (as many of them are indeed assembled in macro-structures as micro-tubules or actin fibers or similar). Could the authors comment on this point?

We agree with the reviewer and this is indeed a very good point. To highlight the complexity of the cytoplasm and the contribution of macro-structures such as ribosomes, we performed fractionation and add-back experiments using the *Xenopus* egg extract system (Figure 5n and o in the manuscript and attached below). We show that the combined effect of adding ribosomes and glycogen helps recover NC densities to physiological ranges. This shows that heavy components add significantly to the dry mass but are neglectable for the osmotic pressure.

Heavy, osmotically inactive components make the cytoplasm denser than the nucleus

(a) Representative images of nuclei assembled in low-speed (LS), high-speed (HS) or high-speed supplemented with purified ribosomes alone (HS +R) or both ribosomes & glycogen (HS + R + G). Panel on the left shows DNA stained with Hoechst-33342 and the panel on the right shows the RI image. Boxes on the right show the RI range. Scale bar: 5 μm.

(b) Quantification of NC density ratio for nuclei reconstituted in each condition. Each symbol represents the value from one nucleus (n = 182 for LS, 30 for HS, 10 for HS + R and 30 for HS + R +G). Black dashed line shows the cytoplasmic density. Black lines and bars show the Mean ± SEM. Mann Whitney test was used to test for statistical significance, where **** indicates p < 0.0001 and ns indicates p > 0.05.

Specifically, the formation of cytoskeletal macrostructures and their impact on colloid osmotic pressure is an interesting question. To test this, we performed nuclear assembly reactions in extract containing the microtubule polymerisation inhibitor Nocodazole and found no difference in nuclear volume, dry mass or density (see data below). Regarding actin networks, we would like to remind the reviewers that the *Xenopus* extract is commonly prepared using Cytochalasin D, an inhibitor of actin polymerisation. Certainly, in a cellular context the formation of cytoskeletal filaments on the pressure balance could be an important factor.

Microtubule depolymerisation does not influence nuclear size or density

(a) Representative RI images of nuclei reconstituted in control extract (Con) or extract containing 10 μ M Nocodazole (Noc). Bar on the right shows the RI range. Scale bar: 5 μ m. **(b-d)** Quantification of nuclear volume (V_n), nuclear dry mass (M_n) and density (ρ_n). Each symbol represents the value from a single nucleus ($n = 10$ nuclei). Black lines and bars show the mean \pm SEM. Mann Whitney test was used to test for statistical significance, where ns indicates $p > 0.05$.

References

- Connelly, K. E., Hedrick, V., Paschoal Sobreira, T. J., Dykhuizen, E. C., & Aryal, U. K. (2018). Analysis of human nuclear protein complexes by quantitative mass spectrometry profiling. *Proteomics*, 18(11), 1700427.
- Dekker, J., & van Steensel, B. (2013). The spatial architecture of chromosomes. In *Handbook of Systems Biology* (pp. 137-151). Academic Press.
- Drew, K., Wallingford, J. B., & Marcotte, E. M. (2021). hu. MAP 2.0: integration of over 15,000 proteomic experiments builds a global compendium of human multiprotein assemblies. *Molecular systems biology*, 17(5), e10016.
- Finan, J. D., Chalut, K. J., Wax, A., & Guilak, F. (2009). Nonlinear osmotic properties of the cell nucleus. *Annals of biomedical engineering*, 37, 477-491.
- Frey, S., Rees, R., Schünemann, J., Ng, S. C., Fünfgeld, K., Huyton, T., & Görlich, D. (2018). Surface properties determining passage rates of proteins through nuclear pores. *Cell*, 174(1), 202-217.
- Guy, A. L., & Taylor, J. H. (1978). Actinomycin D inhibits initiation of DNA replication in mammalian cells. *Proceedings of the National Academy of Sciences*, 75(12), 6088-6092.
- Halverson, J. D., Smrek, J., Kremer, K., & Grosberg, A. Y. (2014). From a melt of rings to chromosome territories: the role of topological constraints in genome folding. *Reports on Progress in Physics*, 77(2), 022601.
- Havugimana, P. C., Hart, G. T., Nepusz, T., Yang, H., Turinsky, A. L., Li, Z., ... & Emili, A. (2012). A census of human soluble protein complexes. *Cell*, 150(5), 1068-1081.
- Hellsten, U., Harland, R. M., Gilchrist, M. J., Hendrix, D., Jurka, J., Kapitonov, V., ... & Rokhsar, D. S. (2010). The genome of the Western clawed frog *Xenopus tropicalis*. *Science*, 328(5978), 633-636.
- Ishida, R., Sato, M., Narita, T., Utsumi, K. R., Nishimoto, T., Morita, T., ... & Andoh, T. (1994). Inhibition of DNA topoisomerase II by ICRF-193 induces polyploidization by uncoupling chromosome dynamics from other cell cycle events. *The Journal of cell biology*, 126(6), 1341-1351.
- Kim, D. H., Li, B., Si, F., Phillip, J. M., Wirtz, D., & Sun, S. X. (2016). Volume regulation and shape bifurcation in the cell nucleus. *Journal of cell science*, 129(2), 457-457.
- Kopito, R. B., & Elbaum, M. (2007). Reversibility in nucleocytoplasmic transport. *Proceedings of the National Academy of Sciences*, 104(31), 12743-12748.
- Lemière, J., Real-Calderon, P., Holt, L. J., Fai, T. G., & Chang, F. (2022). Control of nuclear size by osmotic forces in *Schizosaccharomyces pombe*. *Elife*, 11, e76075.
- Macaulay, C., & Forbes, D. J. (1996). Assembly of the nuclear pore: biochemically distinct steps revealed with NEM, GTP gamma S, and BAPTA. *The Journal of cell biology*, 132(1), 5-20.
- Mes-Hartree, M., & Armstrong, J. B. (1976). Lipid composition of developing *Xenopus laevis* embryos. *Canadian Journal of Biochemistry*, 54(6), 578-582.
- Rollin, R., Joanny, J. F., & Sens, P. (2023). Physical basis of the cell size scaling laws. *Elife*, 12, e82490.
- Sugawara, T., & Kimura, A. (2017). Physical properties of the chromosomes and implications for development. *Development, growth & differentiation*, 59(5), 405-414.
- Webster, M., Witkin, K. L., & Cohen-Fix, O. (2009). Sizing up the nucleus: nuclear shape, size and nuclear-envelope assembly. *Journal of cell science*, 122(10), 1477-1486.
- Wühr, M., Güttler, T., Peshkin, L., McAlister, G. C., Sonnett, M., Ishihara, K., ... & Gygi, S. P. (2015). The nuclear proteome of a vertebrate. *Current biology*, 25(20), 2663-2671.

Reviewer #1 (Remarks to the Author):

This further revised manuscript satisfies my few remaining minor concerns. In my opinion it is ready for publication

Thank you.

Reviewer #4 (Remarks to the Author):

I am happy with the revisions, and I think the manuscript should be accepted. I would only encourage the authors to use a bit more cautionary sentences in main text and SI concerning the model conclusions. They addressed all the points raised, but parameters remain an issue and many of the falsifications arguments they produced rely on quantitative discrepancies that are not very large, and could be parameter-dependent. The debate on such models remains open and while the authors' contribution is very valuable, it is likely new data will help us draw a clearer picture in the coming years: in this perspective it is in the interest of everyone to formulate robust statements about current results.

We thank reviewer #4 for this note. We have included the following statement, lines 366 -372 of the main text: Nonetheless, it is crucial to recognize that the system parameters can significantly vary across different biological systems, e.g. cytoplasmic and nuclear densities as shown in Fig. 1 and this variation also carries over to the pressure values (see section 1.1.4 in Supplementary Note - Theory). Therefore, an accurate, system-specific accounting of the biophysical parameters will be an essential component in future modeling efforts together with developing more advanced models pertinent to the biological system in question.

In addition, we changed / extended the following sections in the Supplementary Theory part of the manuscript:

- In section 1.1.4, the following sentence:

“The reason for such a difference is twofold.”

was extended to:

“The reason for such a difference is twofold and highlights the importance of using accurate, system-specific parameters in model calculations.”

- In section 1.1.4, the following sentence was added:

“The different parameter values corresponding to the *X. laevis* and human systems, can lead to order of magnitude differences in pressure estimates within the same model.”

- In section 1.1.4, the following sentence:

“Therefore, it is important to use accurate values for the biophysical parameters of the system being considered.”

was changed to:

“Therefore, it is important to use accurate, system specific values for the biophysical parameters of the system being considered.”

- Equations (9)-(19) were modified to the desired format.

Reviewer #5 (Remarks to the Author):

The authors made a significant effort to address my concerns (as well as those of the other reviewers), which is greatly appreciated.

They did an excellent job in thoroughly answering and discussing the raised questions.

I therefore fully endorse the manuscript for publication.

Thank you.